# The transcription factor Xrp1 is required for PERK-mediated antioxidant gene induction in *Drosophila*

Brian Brown[†], Sahana Mitra[†], Finnegan D Roach, Deepika Vasudevan, Hyung Don Ryoo*

NYU Grossman School of Medicine, New York, United States

**Abstract** PERK is an endoplasmic reticulum (ER) transmembrane sensor that phosphorylates eIF2α to initiate the Unfolded Protein Response (UPR). eIF2α phosphorylation promotes stress-responsive gene expression most notably through the transcription factor ATF4 that contains a regulatory 5' leader. Possible PERK effectors other than ATF4 remain poorly understood. Here, we report that the bZIP transcription factor Xrp1 is required for ATF4-independent PERK signaling. Cell-type-specific gene expression profiling in *Drosophila* indicated that delta-family glutathione-S-transferases (*gstD*) are prominently induced by the UPR-activating transgene *Rh1*$^{G69D}$. *Perk* was necessary and sufficient for such *gstD* induction, but *ATF4* was not required. Instead, *Perk* and other regulators of eIF2α phosphorylation regulated Xrp1 protein levels to induce *gstD*s. The *Xrp1* 5' leader has a conserved upstream Open Reading Frame (uORF) analogous to those that regulate *ATF4* translation. The *gstD-GFP* reporter induction required putative Xrp1 binding sites. These results indicate that antioxidant genes are highly induced by a previously unrecognized UPR signaling axis consisting of PERK and Xrp1.

*For correspondence:
hyungdon.ryoo@nyumc.org

[†]These authors contributed equally to this work

Competing interest: The authors declare that no competing interests exist.

## Introduction

The endoplasmic reticulum (ER) is the site where most membrane and secretory proteins undergo folding and maturation. This organelle contains an elaborate network of chaperones, redox buffers, and signaling mediators, which work together to maintain ER homeostasis. When the amount of misfolded or nascent proteins exceeds the folding capacity of a given cell, the ER initiates a gene expression regulatory program that is referred to as the Unfolded Protein Response (UPR) (*Karagöz et al., 2019*; *Walter and Ron, 2011*).

The ER also represents an important nexus between protein folding and oxidative stress. The ER maintains an oxidizing environment for the formation of intra- and intermolecular disulfide bonds that contribute to the oxidative folding of client proteins. A product of this reaction is hydrogen peroxide (*Gross et al., 2006*; *Tu and Weissman, 2004*; *Tu and Weissman, 2002*), and excessive protein misfolding in the ER can cause the accumulation of reactive oxygen species (ROS). Consistently, genes involved in redox homeostasis are induced in response to ER stress (*Cullinan et al., 2003*; *Harding et al., 2003*; *Malhotra et al., 2008*; *Mukaigasa et al., 2018*).

In metazoans, there are three evolutionarily conserved branches of the UPR initiated by the ER transmembrane proteins IRE1, PERK (PKR-like ER Kinase, also known as Pancreatic ER Kinase (PEK)), and ATF6 (*Walter and Ron, 2011*). The best studied downstream effectors of IRE1 and PERK signaling are the bZIP family transcription factors XBP1 and ATF4, respectively. Once activated in response to ER stress, these transcription factors induce the expression of genes involved in ER quality control, antioxidant response, and amino acid transport (*Han et al., 2013*; *Harding et al., 2003*; *Walter and Ron, 2011*). The *Drosophila* genome encodes mediators of all three branches of the UPR, and the

roles of the IRE1-XBP1 and PERK-ATF4 branches in *Drosophila* development and tissue homeostasis have been established (*Mitra and Ryoo, 2019*; *Ryoo, 2015*).

The PERK branch of UPR draws considerable interest in part because its abnormal regulation underlies many metabolic and neurodegenerative diseases (*Delépine et al., 2000*; *Ma et al., 2013*; *Pennuto et al., 2008*). Stress-activated PERK is best known to initiate downstream signaling by phospho-inhibiting the translation initiation factor eIF2α (*Harding et al., 1999*; *Shi et al., 1998*). While most mRNA translation becomes attenuated under these conditions, ATF4 protein synthesis increases to mediate a signaling response. Such ATF4 induction requires ATF4's regulatory 5' leader sequence that has an upstream Open Reading Frame (uORF) that overlaps with the main ORF in a different reading frame. This overlapping uORF interferes with the main ORF translation in unstressed cells. (*Harding et al., 2000*; *Kang et al., 2015*; *Vattem and Wek, 2004*). But eIF2α phosphorylation causes the scanning ribosomes to bypass this uORF, ultimately allowing the translation of the main ORF assisted by the noncanonical translation initiation factors eIF2D and DENR (*Bohlen et al., 2020*; *Vasudevan et al., 2020*). The literature also reports PERK effectors that may be independent of ATF4. These include a small number of factors that are translationally induced in parallel to ATF4 in stressed mammalian cells (*Andreev et al., 2015*; *Baird et al., 2014*; *Palam et al., 2011*; *Zhou et al., 2008*). Compared to the ATF4 axis, the roles of these ATF4-independent PERK effectors remain poorly understood.

Here, we report that a previously uncharacterized UPR signaling axis is required for the expression of the most significantly induced UPR targets in the larval eye disc of *Drosophila melanogaster*. Specifically, glutathione-S-transferases (*gstD*s) were among the most significantly induced UPR target genes in *Drosophila*. We further show that such *gstD* induction was dependent on *Perk*, but did not require *crc*, the *Drosophila* ortholog of ATF4. Instead, this response required *Xrp1*, which encodes a bZIP transcription factor with no previously established connections to the UPR. Together, these findings suggest that PERK-Xrp1 forms a previously unrecognized signaling axis that mediates the induction of the most highly upregulated UPR targets in *Drosophila*.

## Results

### RNA-sequencing reveals upregulation of antioxidant genes in response to ER stress

The *Drosophila* eye imaginal disc has been used as a platform to study ER stress responses (*Kang and Ryoo, 2009*; *Kang et al., 2012*; *Ryoo et al., 2007*), but the global transcriptional changes associated with ER stress have not been thoroughly investigated in this context. To probe these changes in an unbiased approach, we performed RNA-seq on control and ER-stressed third instar eye imaginal discs. To genetically impose ER stress in this tissue, we misexpressed a mutant allele of Rhodopsin-1 (Rh1), a multipass membrane protein that is synthesized in the ER (*O'Tousa et al., 1985*; *Zuker et al., 1985*). The mutant allele *Rh1^{G69D}* has been used as a potent genetic inducer of ER stress and UPR signaling (*Colley et al., 1995*; *Kang and Ryoo, 2009*; *Kurada and O'Tousa, 1995*; *Ryoo et al., 2007*). We chose to drive the expression of *Rh1^{G69D}* in eye discs posterior to the morphogenetic furrow using Gal4 driven by the synthetic Glass Multiple Repeat promoter (henceforth referred to as *GMR > Rh1^{G69D}*) (*Brand and Perrimon, 1993*; *Freeman, 1996*). To profile gene expression specifically in these cells, we simultaneously drove the expression of nuclear envelope-localized *EGFP::Msp300^{KASH}* (*Hall et al., 2017*; *Ma and Weake, 2014*), enabling us to isolate only EGFP-labeled nuclei for further analysis (*Figure 1A*) (see Materials and methods). We confirmed successful enrichment of EGFP expressing nuclei through qRT-PCR and imaging (*Figure 1B and C*), and prepared total RNA from them for RNA-seq. The resulting RNA-seq data was subjected to differential expression analysis to identify genes induced by ER stress (*Supplementary file 1*). The full results of our RNA-seq experiment are available through the Gene Expression Omnibus (GEO; accession number GSE150058).

Gene Ontology (GO) term analysis of the RNA-seq data indicated that genes related to glutathione metabolism ($p = 2.08*10^{-4}$) and cell redox homeostasis ($p = 3.29*10^{-6}$) were noticeably enriched in *GMR > Rh1^{G69D}* samples. Among this group, the most significantly upregulated gene was the delta family glutathione-S-transferase *gstD1* (*Figure 1D*). Other notable genes induced by ER stress included the superoxide dismutase *Sod2*, the peroxiredoxin *Jafrac1* as well as multiple other glutathione-S-transferase genes. These findings were suggestive of an antioxidant response. Consistently, we found

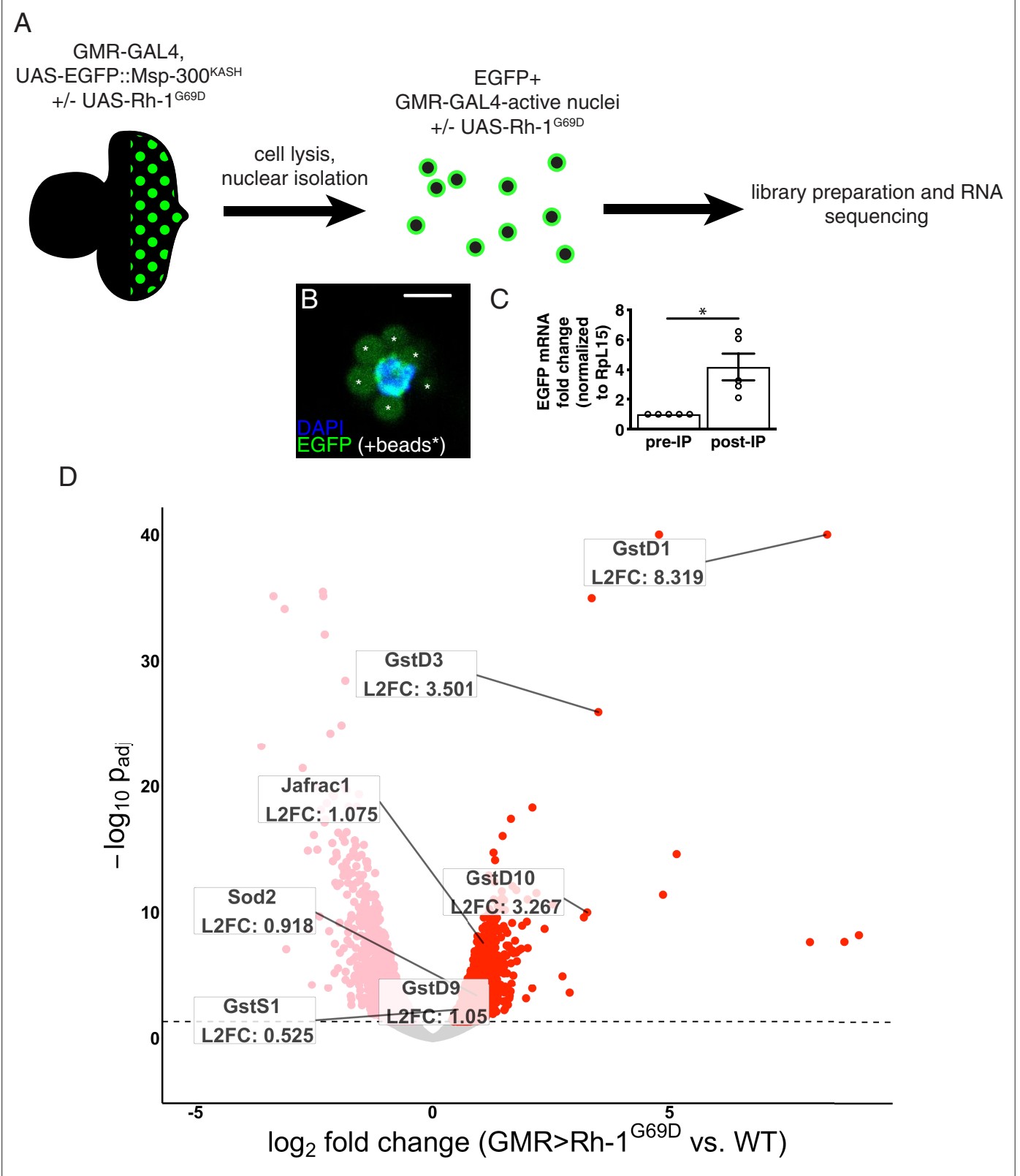

**Figure 1.** *RNA-sequencing reveals upregulation of antioxidant genes in response to ER stress.* (**A**) Workflow for isolation of tagged nuclei from eye imaginal discs and preparation of total RNA for transcriptional profiling. (**B**) A representative image of an isolated, EGFP-tagged nucleus. Asterisks label auto-fluorescent anti-EGFP-coupled beads. Scale bar = 5 μm. (**C**) Representative qRT-PCR of pre- and post-isolation disc nuclei showing a significant enrichment of EGFP mRNA following isolation. The error bar represents standard error (SE). Statistical significance was assessed through a two

*Figure 1 continued on next page*

*Figure 1 continued*

tailed t-test. * = p < 0.05. (**D**) Volcano plot of genes differentially expressed in the presence or absence of Rh1^G69D expression. Genes significantly up/downregulated in response to *Rh1^G69D* are shown in red and pink, respectively. A select set of antioxidant genes induced by *Rh1^G69D* are labeled with their $\log_2$ fold change values. Note: $-\log_{10} p_{adj}$ of *gstD1* exceeded the bounds of the graph (180) and was constrained to the maximum displayed value of 40 for readability.

The online version of this article includes the following figure supplement(s) for figure 1:

**Figure supplement 1.** *Evidence for ROS build up in eye discs expressing Rh1^G69D.*

that *GMR > Rh1^G69D* discs showed signs of ROS accumulation. Specifically, the ROS sensitive reporter cyto-roGFP2-Orp1 that gains fluorescence upon ROS-initiated Orp1 oxidation (*Albrecht et al., 2011*) was activated in *GMR > Rh1^G69D* but not in control eye imaginal discs (*Figure 1—figure supplement 1*). These observations prompted us to investigate the link between ER stress and antioxidant gene expression.

## ER stress induces *gstD-GFP* expression in *Drosophila* imaginal discs

To validate our RNA-seq findings, we utilized a transgenic reporter of *gstD1* expression, *gstD-GFP*, in which a regulatory DNA sequence between the *gstD1* and *gstD2* genes drives the expression of GFP (*Figure 2A*; *Sykiotis and Bohmann, 2008*). While control discs expressing *lacZ* (*GMR > lacZ*) showed no discernible *gstD-GFP* expression in the posterior eye disc (*Figure 2B and D*), we observed marked induction of *gstD-GFP* in *GMR > Rh1^G69D* discs (*Figure 2C and D*), in agreement with our RNA-seq results.

Examination of the *gstD-GFP* reporter also allowed us to visualize the pattern of gene expression in response to *Rh1^G69D* expression. Expression of *GMR-Gal4* driven *Rh1^G69D* expression was detected in all cell types of the posterior eye disc (*Figure 2C and E*), but among this population of cells, *gstD-GFP* expression occurred primarily in non-photoreceptor cells. This was evident in eye discs regions with an apical-to-basal orientation, where *gstD-GFP* signal was primarily detected in the basal support cell layer. GFP was not present in the apical photoreceptor layer marked by anti-Elav antibody labeling, even though those apical cells expressed *Rh1^G69D* (*Figure 2E and F*). Confocal sections on the apical layer with photoreceptor cell bodies further confirmed that the *gstD-GFP*-positive signal was coming from non-photoreceptor cells (*Figure 2G*). We therefore conclude that *gstD-GFP* induction by ER stress occurs in a cell-type-specific manner.

To further validate that *gstD-GFP* is induced by ER stress, we treated imaginal discs with culture media containing dithiothreitol (DTT), which causes ER stress by interfering with oxidative protein folding. Most cells of the larval eye and wing imaginal discs did not express *gstD-GFP* under control conditions. However, 2 mM DTT treatment for 4 hr induced *gstD-GFP* expression in a large population of cells in both the eye and wing imaginal discs (*Figure 2—figure supplement 1*). Similar to the outcome of *Rh1^G69D* overexpression, DTT treatment induced *gstD-GFP* primarily in the non-neuronal cell layer (*Figure 2—figure supplement 1* A-D). *gstD-GFP* expression was also induced by a different ER stress-imposing chemical, the *N*-linked glycosylation inhibitor tunicamycin (*Figure 2—figure supplement 1* G-H). These results further support the notion that *gstD*s are significant UPR targets in *Drosophila* tissues, with some degree of cell-type specificity.

## ER stress-induced *gstD-GFP* expression does not require the antioxidant response factor *cncC*

The *gstD-GFP* reporter responds to known oxidative stressors, including hydrogen peroxide and the oxygen radical generator N,N'-dimethyl-4,4'-bipyridinium dichloride (paraquat) (*Sykiotis and Bohmann, 2008*). The transcription factor Cap'n'collar-C (cncC), an Nrf2 homolog, mediates this response in *Drosophila* (*Hochmuth et al., 2011*; *Sykiotis and Bohmann, 2008*). This led us to test whether *cncC* is required for ER stress-mediated *gstD-GFP* expression. Specifically, we generated mutant mitotic clones negatively marked with *armadillo-lacZ* in *GMR > Rh1^G69D* eye discs. We examined two different mutant alleles of *cnc*; the *vl110* allele which has a deletion that spans significant parts of the coding sequence (*Mohler et al., 1995*), and the *k6* allele which has a nonsense mutation that specifically truncates longer ROS-responsive Cnc isoforms including CncC (*Veraksa et al., 2000*). Mutant mosaic clones of these *cncC* alleles failed to block the induction of *gstD-GFP* by *GMR >*

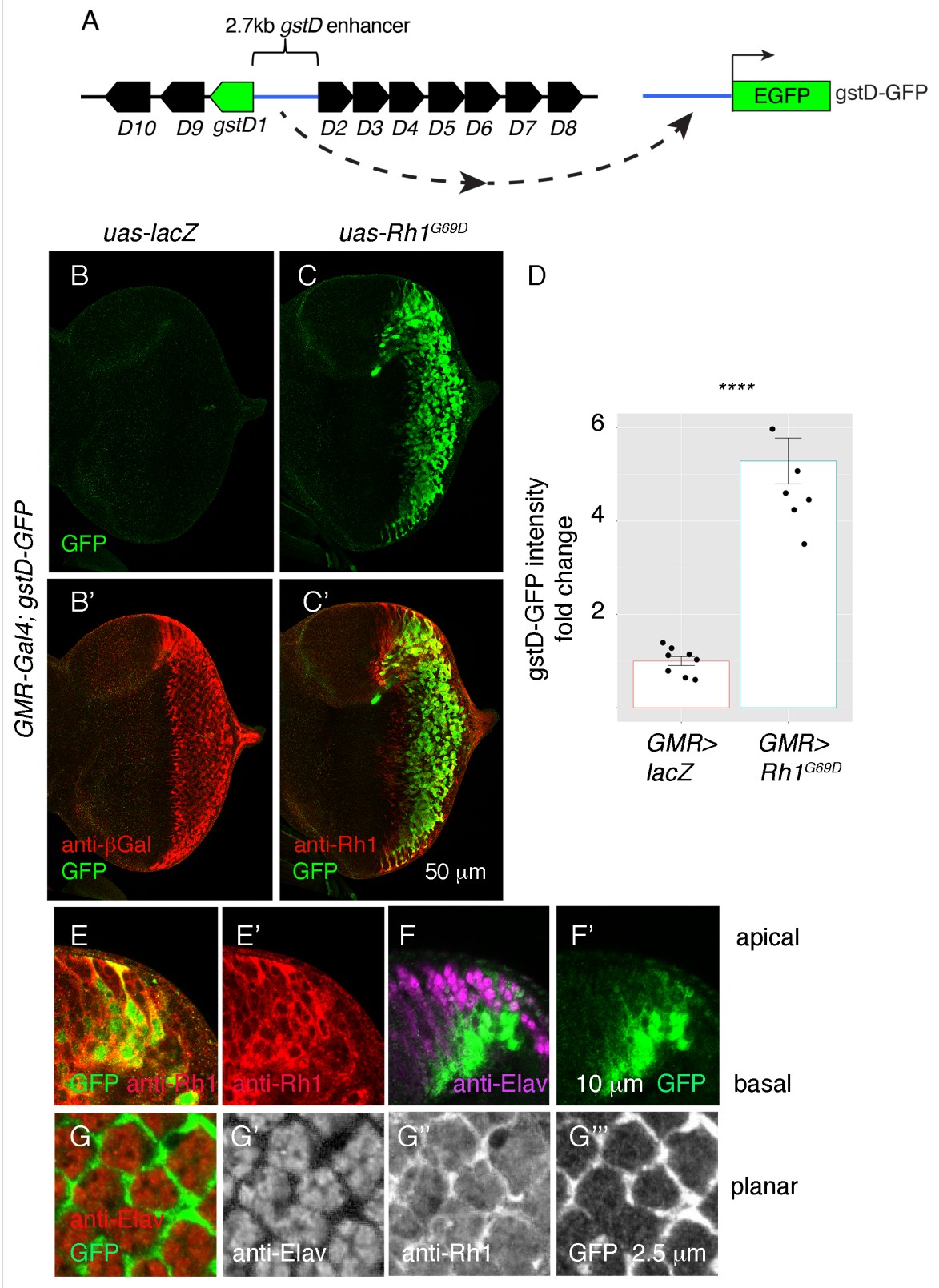

**Figure 2.** *ER stress induces gstD-GFP expression in the eye imaginal disc.* (**A**) Schematic of the *gstD-GFP* reporter and the *gstD* locus, which encodes multiple *gstD* isozymes in close proximity to the 2.7 kb *gstD1/gstD2* intergenic enhancer (scale is approximate). (**B–C**) Planar views of *gstD-GFP* larval eye discs also expressing either *lacZ* or *Rh1^G69D* driven by *GMR-Gal4*. (**B', C'**) show merged images of discs with *gstD-GFP* expression in green, and Rh1 or lacZ in red. (**B, C**) show only the green channel. Anterior is to the left and posterior to the right. Scale bar = 50 μm. (**D**) Quantification of gstD-GFP

*Figure 2 continued on next page*

Figure 2 continued
pixel intensity fold change from eye discs with the indicated genotypes. Statistical significance based on t test (two tailed). **** = p < 0.0001. (E, F) A magnified view of *GMR > Rh1^G69D* eye discs in apico-basal orientation. gstD-GFP signals are marked in green. (E) Posterior eye disc double labeled with anti-Rh1 (red). (F) An equivalent region labeled with anti-Elav antibody that marks photoreceptors (magenta). Scale bar = 10 μm (G) A magnified view of *GMR > Rh1^G69D* eye discs in planar orientation with gstD-GFP (green) and anti-Elav labeling (red). Individual channels are shown separately in (G' – G'''). Scale bar = 2.5 μm.

The online version of this article includes the following figure supplement(s) for figure 2:

**Figure supplement 1.** *gstD-GFP induction by ER-stress causing chemicals.*

**Figure supplement 2.** *ER stress-induced expression of gstDs is independent of cncC.*

**Figure supplement 3.** *Rh1^G69D -induced expression of gstD-GFP neither requires Ire1 nor Atf6.*

*Rh1^G69D*. Contrary to our expectations, *cncC* mutant clones showed enhanced *gstD-GFP* expression as compared to the neighboring wild-type cells (*Figure 2—figure supplement 2* A-C). This enhanced *gstD-GFP* expression in the mutant clones could reflect increased proteostatic stress associated with *cncC* loss. Based on these data, we conclude that *cncC* is not required for *gstD-GFP* induction in response to ER stress.

Consistent with these genetic experiments, we also observed that transcriptional induction of *gstD*s in cultured *Drosophila* S2 cells was *cncC*-independent. When ER stress was pharmacologically induced in *Drosophila* S2 cells by tunicamycin, we observed robust transcriptional induction of *gstD2*, which shares the same 2.7 kb enhancer as *gstD1* (*Tang and Tu, 1994*; *Toung et al., 1993*). Such induction of *gstD2* was unaffected when *cncC* was knocked down in S2 cells prior to tunicamycin treatment (*Figure 2—figure supplement 2* D, E). The knockdown efficiency of *cnc* as estimated through q-RT-PCR from these cells was 92.46 %. As a control, we validated the known roles of *cncC* in antioxidant response by utilizing the oxidative stressor paraquat, which leads to transcriptional induction of *gstD2* in S2 cells as well (*Figure 2—figure supplement 2* E). Consistent with previous reports, induction of *gstD2* in response to paraquat was blocked after *cncC* knockdown in S2 cells (*Figure 2—figure supplement 2* E). These results further support the idea that *cncC* is not required for tunicamycin-induced *gstD* gene expression.

## gstD-GFP induction by ER stress requires PERK that phosphorylates eIF2α

We next considered the three canonical branches of the UPR as candidate mediators of *gstD-GFP* induction by *GMR > Rh1^G69D*. We found no evidence for the requirement of *Ire1* or *Atf6*. Specifically, *GMR > Rh1^G69D* eye discs with negatively marked *Ire1* mutant mosaic clones still induced *gstD-GFP*. If anything, there appeared to be a general increase in *gstD-GFP* expression in *Ire1* mutant clones (*Figure 2—figure supplement 3* A), consistent with previous observations that impairment of the IRE1 axis of the UPR could cause a concomitant increase in the activity of the remaining UPR branches (*Huang et al., 2017*). Likewise, *Atf6* mutant discs bearing a piggyBac insertion in the *Atf6* coding sequence, *Atf6^LL07432*, did not impair *gstD-GFP* induction by *GMR > Rh1^G69D* (*Figure 2—figure supplement 3* B, C).

We next tested whether *gstD-GFP* induction was reliant on *Perk* using the previously described *Perk^e01744* mutant allele (*Wang et al., 2015*). In contrast to *Atf6* mutant discs and *Ire1* mutant clones, *Perk* loss-of-function mutant eye discs expressing *GMR > Rh1^G69D* were unable to induce *gstD-GFP* (*Figure 3A–C and E*). Conversely, we overexpressed *Perk* in eye discs. We had previously shown that such overexpression of *Perk* is sufficient for its autoactivation, possibly by overwhelming the repressive capacity of BiP (*Malzer et al., 2010*). Accordingly, we found that overexpression of *Perk* was sufficient to induce *gstD-GFP* expression (*Figure 3D and E*).

PERK is best known to phosphorylate eIF2α in response to ER stress (*Figure 3—figure supplement 1* A). To test if such kinase activity is required for *gstD-GFP* induction, we expressed a *Perk* transgene with a mutation that disrupts its kinase activity (*Malzer et al., 2010*). While wild-type *Perk* expression robustly induced *gstD-GFP*, the *kinase dead Perk* (*Perk^KD*) transgene failed to induce the reporter under otherwise equivalent conditions (*Figure 3—figure supplement 1* B, C). To further test if eIF2α phosphorylation causes the induction of *gstD-GFP*, we employed an RNAi line against *gadd34* (*Malzer et al., 2010*). *gadd34* encodes a phosphatase subunit that helps to dephosphorylate

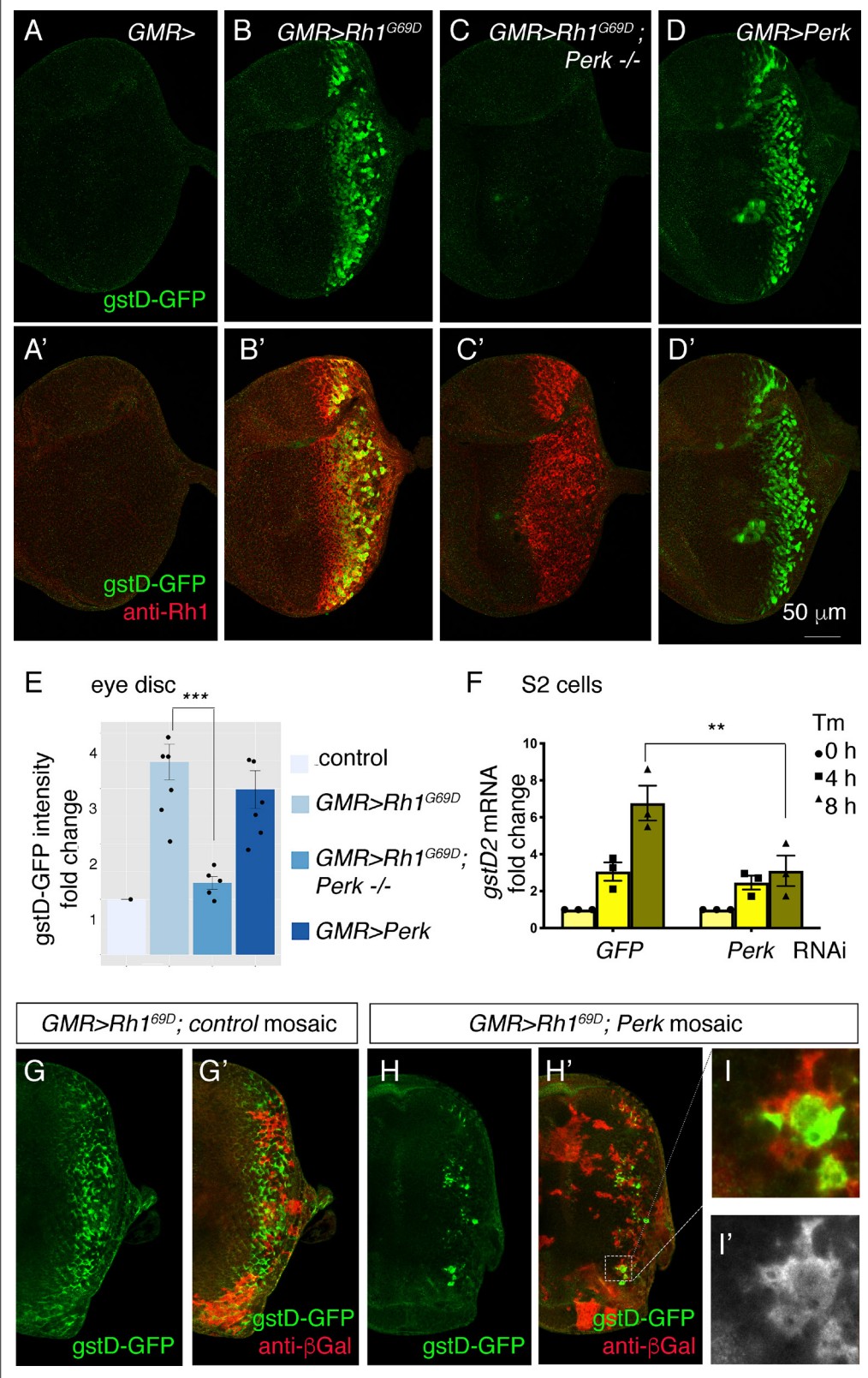

**Figure 3.** *Perk is necessary and sufficient to induce gstD-GFP expression.* (**A–D**) Representative eye imaginal discs with gstD-GFP (green) alone channels (**A–D**), and gstD-GFP (green) channels merged with anti-Rh1 (red) (**A'-D'**). The genotypes of the discs are as follows: (**A**) *GMR-Gal4; gstD-GFP/+*, (**B**) *GMR-Gal4; gstD-GFP/UAS-Rh1^{G69D}*, (**C**) *GMR-Gal4; gstD-GFP/UAS-Rh1^{G69D}; Perk^{e01744}* (**D**) *GMR-Gal4; gstD-GFP/UAS-Perk.* (**E**) Quantification of gstD-GFP

*Figure 3 continued on next page*

*Figure 3 continued*

pixel intensity fold change from posterior eye discs of the indicated genotypes from A-D. (**F**) qRT-PCR of *gstD2* normalized to RpL15 in S2 cells treated with 10 ug/mL tunicamycin for 0, 4, or 8 hr. Cells were either pre-treated with control dsRNA (GFP) or those targeting *Perk*. (**G–I**) *gstD-GFP* (green) containing *GMR > Rh1$^{G69D}$* eye discs with either control mosaic clones (**G**), or those with *Perk* loss-of-function clones (**H, I**). Homozygous *Perk$^{e01744}$* clones in H' and I are marked by the absence of β-galactosidase (red) expression. Note that gstD-GFP expression occurs broadly in the background of control mosaic clones, but is largely absent in *Perk* homozygous mutant clones. (**I, I'**) A magnified view of the dotted inset in (**H'**). (**I'**) β-galactosidase only channel (marking *Perk*-positive cells; in white). Scale bar = 50 μm. * = p < 0.05, ** = p < 0.005, *** = p < 0.001.

The online version of this article includes the following figure supplement(s) for figure 3:

**Figure supplement 1.** PERK's kinase domain and eIF2α phosphorylation mediate *gstD-GFP* induction.

eIF2α, and therefore, the *loss of gadd34* increases phospho-eIF2α levels (*Figure 3—figure supplement 1* A). We found that knockdown of *gadd34* using the eye specific *GMR-Gal4* and *ey-Gal4* drivers induces *gstD-GFP* in posterior eye discs (*Figure 3—figure supplement 1* D, E). A striking induction of *gstD-GFP* was also observed when *gadd34* was knocked down in the wing disc posterior compartment using the *hh-Gal4* driver (*Figure 3—figure supplement 1* F, G). These results support the idea that PERK-mediated phosphorylation of eIF2α promotes *gstD-GFP* induction across many tissue types.

To independently validate the role of *Perk*, we turned to S2 culture cells where tunicamycin (Tm) treatment induces *gstD2* transcripts as detected through q-RT PCR. *Perk* dsRNA treatment resulted in an 82.7 % reduction in *Perk* transcripts as assessed by q-RT-PCR. Knockdown of *Perk* in this setup blunted the induction of *gstD2* (*Figure 3F*).

To further test if *Perk* has a cell-autonomous role in *gstD-GFP* induction, we examined eye discs with *Perk* loss-of-function mosaic clones. Whereas *GMR> Rh1$^{G69D}$* robustly induced *gstD-GFP* in control mosaic clones without the *Perk* mutation (*Figure 3G*), the *gstD-GFP* signal was largely lost in eye discs with *Perk* homozygous mutant clones (*Figure 3H, I*). The small number of the residual GFP signals came from *Perk + * mosaic clones, indicating that *Perk's* role in *gstD-GFP* expression is cell-autonomous (*Figure 3I*). Together, these results independently support *Perk's* role in *gstD* induction in ER-stressed cells.

## *gstD-GFP* induction does not require *crc*

As ATF4 is the best-characterized downstream effector of PERK in *Drosophila* and mammals (*Mitra and Ryoo, 2019*; *Ryoo, 2015*; *Walter and Ron, 2011*; *Figure 4A*), we examined whether the *Drosophila* ATF4 ortholog *crc* is required for *gstD-GFP* induction in *GMR > Rh1$^{G69D}$* discs. For this, we employed the *crc$^1$* allele, a strong hypomorph bearing a missense mutation in the *crc* coding sequence (*Hewes et al., 2000*). To our surprise, loss of *crc* did not impair *gstD-GFP* induction in the eye discs (*Figure 4B' and C'*). *Rh1$^{G69D}$* expression levels were similar in discs with or without *crc* (*Figure 4B" and C"*). As a positive control to validate *crc* activation, we used *Thor-lacZ*, a *lacZ*-based enhancer trap inserted upstream of the *crc* target gene *Thor* (*Bernal and Kimbrell, 2000*; *Kang et al., 2017*; *Figure 4A*). Consistent with previously published results, *Thor-lacZ* was induced in response to *GMR > Rh1$^{G69D}$* and such induction was suppressed in *crc$^1$* homozygotes (*Figure 4B and C*). In an S2 cell-based assay, knockdown of *crc* did not impair *gstD2* induction in response to tunicamycin treatment (*Figure 4D*), but did block the induction of *Thor* (*Figure 4E*). Consistently, overexpression of *crc* using *UAS-crc$^{leaderless}$* (*Vasudevan et al., 2020*) also did not induce *gstD-GFP* expression (*Figure 4G*). We confirmed that the *uas-crc$^{leaderless}$* transgene is functional, as its expression was sufficient to induce *Thor-lacZ* reporter expression in eye discs (*Figure 4H, I*). These data strongly indicate that the induction of *gstD*s in response to ER stress is dependent on *Perk*, but not *crc*.

## The induction of *gstD* genes and other antioxidants in response to ER stress require the bZIP transcription factor Xrp1

We next sought to determine how PERK activation induced the expression of *gstD-GFP* independently of *crc*, and turned our focus to another stress response transcription factor, *Xrp1*. Although *Xrp1* has no known association with the UPR, *Xrp1* drew our attention as it reportedly mediates the induction of the *gstD1-lacZ* reporter in response to ribosome mutations that cause cell competition (*Ji et al., 2019*).

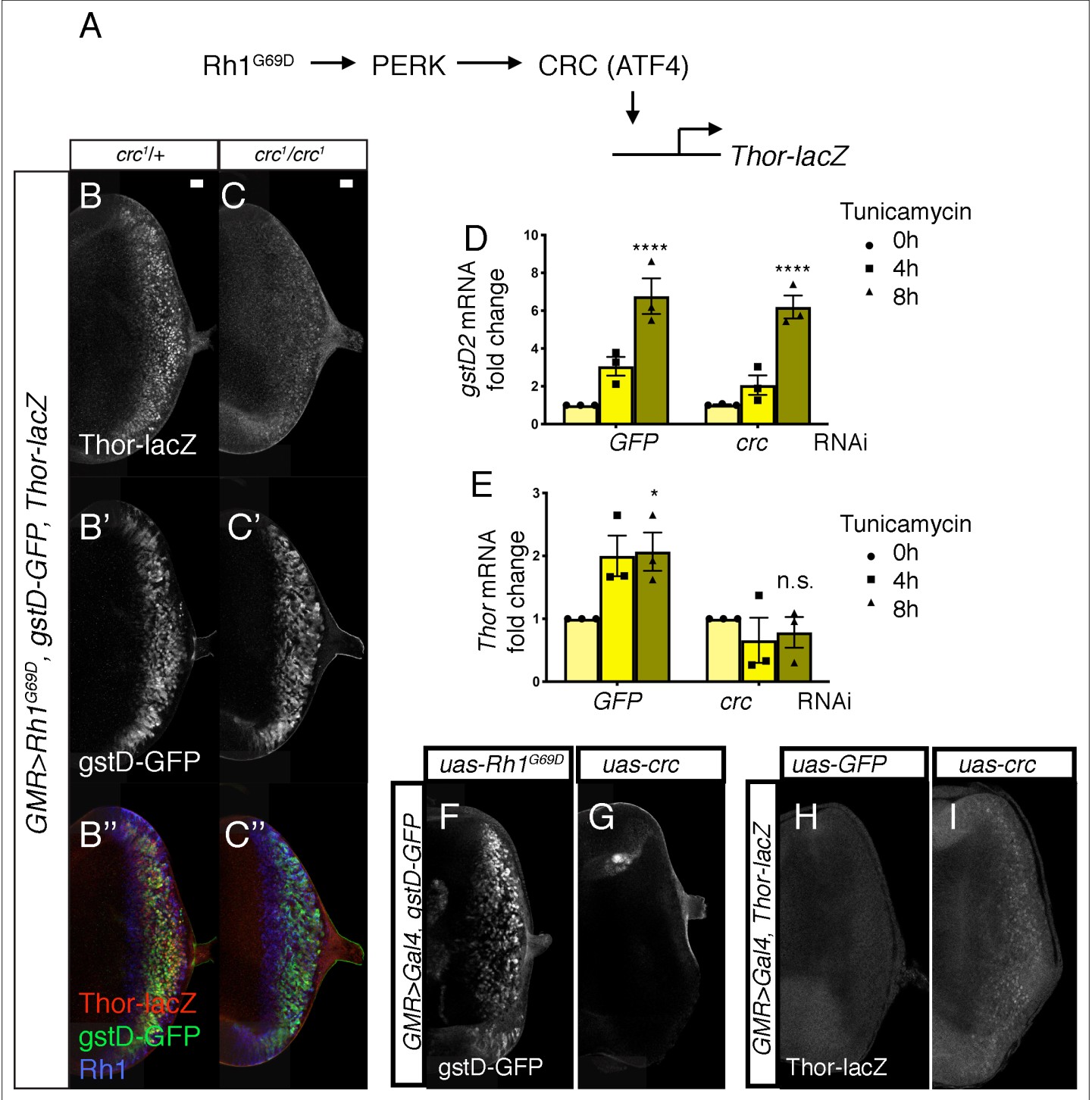

**Figure 4.** *PERK-mediated gstD expression is ATF4 (crc)-independent.* (**A**) Schematic diagram of the *Drosophila* PERK-ATF4 signaling pathway. CRC is the *Drosophila* ATF4 ortholog, and *Thor-lacZ* is a CRC transcriptional target. (**B, C**) Eye imaginal discs expressing *GMR > Rh1*$^{G69D}$ in either the *crc*$^1$/+ control (**B**) or in the *crc*$^1$ homozygous backgrounds (**C**). These discs also have *Thor-lacZ* that reports CRC activity (B, C, in white), and *gstD-GFP* (B', C', in white). (**B", C"**) Images of discs with *Thor-lacZ* (red), *gstD-GFP* (green) and anti-Rh1 (blue) channels merged. Note that *crc*$^1$ homozygous mutants block *Thor-lacZ* expression (**C**), but still allows *gstD-GFP* expression in *Rh1*$^{G69D}$ expressing cells (**C', C"**). (**D, E**) qRT-PCR analysis of *gstD2* and *Thor* normalized to *Rpl15* in S2 cells challenged with 10 μg/mL tunicamycin for 0, 4, or 8 hr. Cells were either treated with control dsRNA (GFP) or that target *crc*. (**F, G**) *gstD-GFP* expression (white) in eye discs that are overexpressing either *Rh1*$^{G69D}$ (**F**) or *crc*$^{leaderless}$ (**G**) through the *GMR-Gal4* driver. (**H, I**) *Thor-lacZ* expression (white) in eye discs overexpressing either a control *GFP* transgene (**H**), or *crc*$^{leaderless}$ (**I**) through the *GMR-Gal4* driver. Scale bar = 20 μm. * = p < 0.05, **** = p < 0.0001. Genotypes: (**B**) *GMR-Gal4; gstD-GFP, crc*$^1$/*UAS-Rh1*$^{G69D}$, *Thor-lacZ*. (**C**) *GMR-Gal4; gstD-GFP, crc*$^1$/*crc*$^1$, *UAS-Rh1*$^{G69D}$, *Thor-*

*Figure 4 continued on next page*

*Figure 4 continued*

*lacZ.* (**F**) *GMR-Gal4; gstD-GFP/UAS-Rh1^{G69D}.* (**G**) *GMR-Gal4; gstD-GFP/UAS-crc^{leaderless}.* (**H**) *GMR-Gal4; Thor-lacZ/UAS-GFP.* (**I**) *GMR-Gal4; Thor-lacZ/UAS-crc^{leaderless}.*

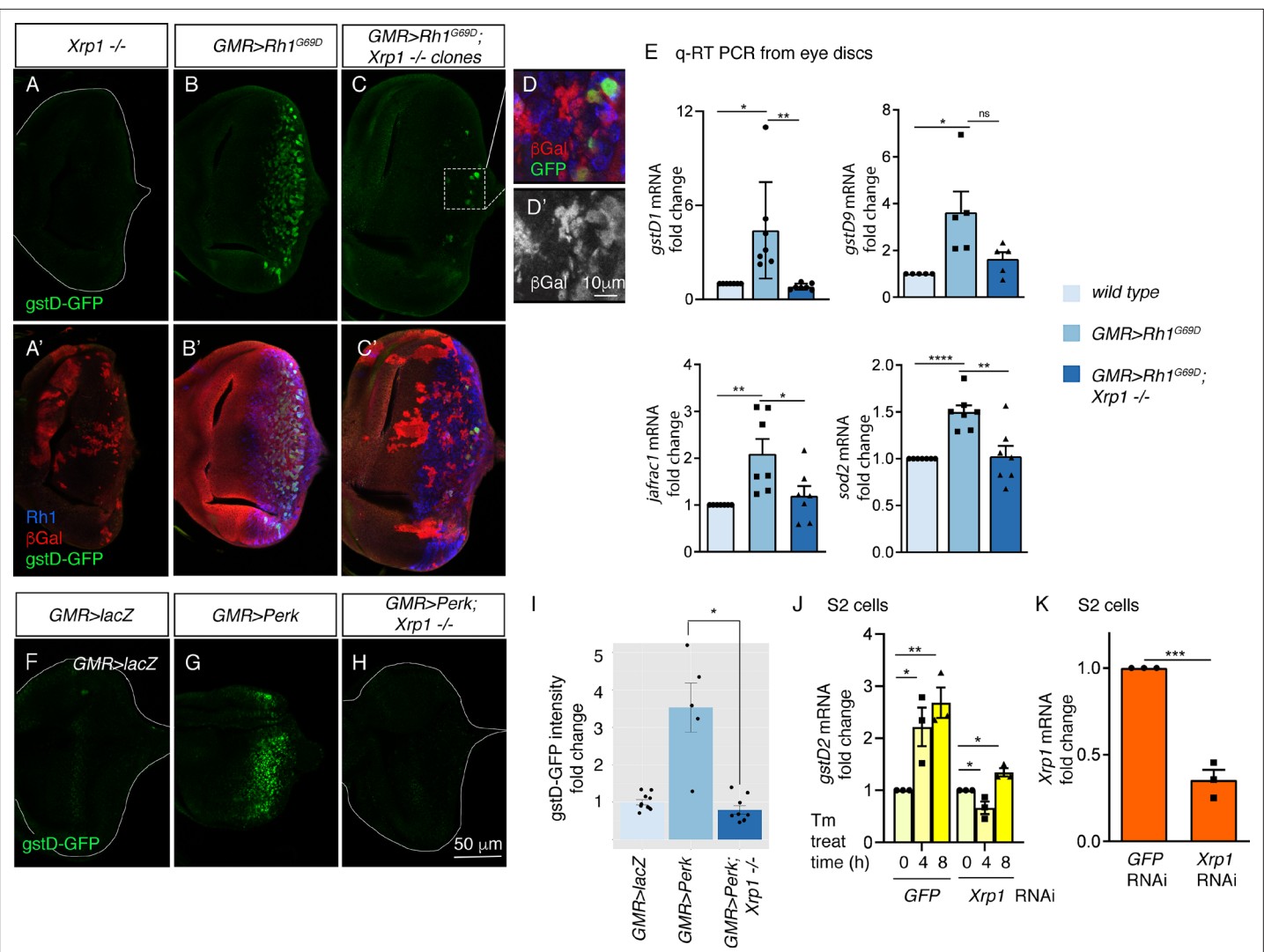

**Figure 5.** *PERK regulates gstD gene expression through the bZIP transcription factor Xrp1.* (**A–G**) gstD-GFP expression (green) in eye discs. (**A, A'**) An eye disc with homozygous *Xrp1^{M2-73}* mutant clones that are marked by the absence of βGal (red). (**B, B'**) A *GMR> Rh1^{G69D}* disc in a control genetic background. (**C, C'**) A *GMR> Rh1^{G69D}* disc with *Xrp1^{M2-73}* loss-of-function mosaic clones (non-red). (**A–C**) gstD-GFP only channels. (**A'-C'**) Merged images with Rh1 is stained in blue and βgal is in red. Note the absence of *gstD-GFP* in *Xrp1* mutant clones. (**D, D'**) Magnified view of the inset marked in C. The scale bar here represents 10 μm. (**D'**) is the βGal only channel of the image in (**D**). (**E**) Quantitative(q) RT-PCR results of indicated genes from dissected eye discs. The genotypes are color labeled. (**F–H**) Eye discs expressing control *lacZ* (**E**), *Perk* (**F**), or expressing *Perk* in the background of *Xrp1^{M2-73}* mosaic clones (**G**). (**I**) Quantification of gstD-GFP intensity fold change in the indicated genotypes. (**J, K**) q RT-PCR results from S2 cells. (**J**) *gstD2* mRNA levels from S2 cells challenged with 10 μg/mL tunicamycin for 0, 4 or 8 hr. The cells were pretreated with dsRNA against either *GFP* (lanes 1–3) or *Xrp1* (lanes 4–6). (**K**) *Xrp1* mRNA levels in cells pretreated with dsRNA against *GFP* or *Xrp1*. All qRT-PCR results were normalized with *Rpl15*. Two tailed t tests were used to assess statistical significance. The scale bar of 50 μm applies to all images except for (**D**). * = p < 0.05, ** = p < 0.005, *** = p < 0.001. Genotypes: (**A**) *GMR-Gal4, ey-FLP; gstD-GFP/+; FRT82, Xrp1^{M2-73}/FRT82, arm-lacZ.* (**B**) *GMR-Gal4, ey-FLP; gstD-GFP/UAS-Rh1^{G69D}.* (**C**) *GMR-Gal4, ey-FLP; gstD-GFP/UAS-Rh1^{G69D}; FRT82, Xrp1^{M2-73}/FRT82, arm-lacZ.* (**E**) *GMR-Gal4, ey-FLP; gstD-GFP/UAS-lacZ.* (**F**) *GMR-Gal4, ey-FLP; gstD-GFP/UAS-Rh1^{G69D}.* (**G**) *GMR-Gal4, ey-FLP; gstD-GFP/UAS-Rh1^{G69D}; FRT82, Xrp1^{M2-73}/ FRT82, arm-lacZ.*

To assess a possible role of *Xrp1* in UPR signaling, we performed clonal analysis with an *Xrp1* mutant allele, *Xrp1^M2-73^*, which bears a nonsense mutation truncating all *Xrp1* isoforms prior to both the AT-hook motif and bZIP domain (*Lee et al., 2018*). Loss of *Xrp1* had no effect on basal *gstD-GFP* expression levels (*Figure 5A*). *GMR > Rh1^G69D^* discs prominently expressed *gstD-GFP*, but otherwise equivalent eye discs with *Xrp1* mutant clones had marked reduction of the *gstD-GFP* signal (*Figure 5B and C*). The small patches of GFP-positive cells in these discs were all within the *Xrp1+* mosaic cells (*Figure 5C'* D).

To validate the findings with the *gstD-GFP* reporter, we assessed the mRNA levels of *gstD1* and other select antioxidant genes from dissected eye imaginal discs through qRT-PCR. A subset of the analyzed transcripts showed induction in *GMR > Rh1^G69D^* disc samples dependent on *Xrp1* (*Figure 5E*). These included *gstD1*, *jafrac1* that encodes a cytoplasmic peroxidase and *superoxide dismutase 2* (*sod2*) that detoxifies superoxides in the mitochondria.

The requirement of *Xrp1* in *gstD-GFP* expression in response to *GMR > Rh1^G69D^* prompted us to further examine the epistatic relationship between *Perk* and *Xrp1*. We found that *gstD-GFP* induction caused by *GMR > Perk* was completely abolished in *Xrp1* mutant eye discs (*Figure 5F–I*). Together, these data indicate that *Xrp1* is epistatic to *Perk* in mediating *gstD-GFP* induction as part of UPR signaling.

We further confirmed the role of *Xrp1* in cultured S2 cells. Tunicamycin treatment induced *gstD2* significantly within 4 and 8 hr, but cells with *Xrp1* knockdown did not show a statistically significant induction (*Figure 5J and K*). Together, these results independently support *Xrp1's* role in *gstD* induction in ER-stressed cells.

## Xrp1 protein is induced by *Perk*-dependent UPR

We next examined if ER stress induces *Xrp1* expression through anti-Xrp1 immunohistochemistry. Control eye imaginal discs did not show obvious anti-Xrp1 signals beyond background (*Figure 6A and A'*), but *GMR > Rh1^G69D^* eye discs displayed nuclear anti-Xrp1 immunoreactivity in cells that expressed Rh1^G69D^ (*Figure 6B, B' and D*). Consistent with the role of *Perk* and eIF2α phosphorylation, Xrp1 protein induction by *GMR > Rh1^G69D^* was significantly reduced in the *Perk^e01744^* mutant background (*Figure 6C, C' and D*). While the reduction was statistically significant (*Figure 6D*), the residual level of Xrp1 in the *Perk* mutant discs indicated a partial dependency of Xrp1 induction on *Perk*. Interestingly, we found that *Xrp1* transcript levels did not change significantly in *GMR > Rh1^G69D^* discs as compared to controls, as assessed from our RNA-seq analysis (*Supplementary file 1*) and by qRT-PCR analysis (*Figure 6E*). These results indicate that *Perk* mediates the induction of Xrp1 protein through a post-transcriptional mechanism.

To test if *Perk's* kinase activity is involved in Xrp1 protein induction, we compared the effects of overexpressing *Perk^WT^* versus *Perk^KD (Kinase Dead)^*. Xrp1 protein levels increased in *GMR > Perk^WT^* eye discs, but not in *GMR > Perk^KD^* discs (*Figure 6F, G and H*). To further test if PERK's phosphorylation target eIF2α controls *gstD-GFP* expression, we knocked down *gadd34* in wing discs using the posterior compartment-specific *hh-Gal4* driver. While control wing discs showed a low basal anti-Xrp1 signal throughout the tissue, depletion of *gadd34* by *RNAi* resulted in distinct nuclear anti-Xrp1 signals throughout the posterior compartment (*Figure 6I and J*). These results indicate that Xrp1 induction is regulated by eIF2α phosphorylation.

PERK's well-established downstream effector, ATF4, has a 5' regulatory leader sequence with multiple upstream Open Reading Frames (uORFs) that allows its translational induction in response to stress. To examine if there is an analogous 5' leader in Xrp1, we used a bioinformatic program that predicts initiation codons (https://atgpr.dbcls.jp). This approach did not detect initiation codons in the 5' leader of Xrp1's shorter splice isoforms (e.g. isoform D and E). By contrast, two putative uORFs in the 5' leader of the long splice isoforms of Xrp1 (e.g. isoform F and G) were identified, with uORF1 predicted to encode a 124 a.a. length peptide, and the uORF2 with a 288 a.a. peptide. The uORF2 overlapped with the main ORF, but was in a different reading frame (*Figure 7A*). Such an arrangement is similar to that of ATF4's last uORF.

To assess the likelihood that Xrp1 uORFs are peptide coding sequences, we performed Protein BLAST searches with the encoded peptide sequences. Xrp1's uORF1 did not have any homologous hits in other species. However, sequences homologous to *D. melanogaster* Xrp1 uORF2 were identified in *D. kikkawai* (Percent identity = 71.85%), *D. persimilis* (Percent identity = 48.67%), *D. navojoa*

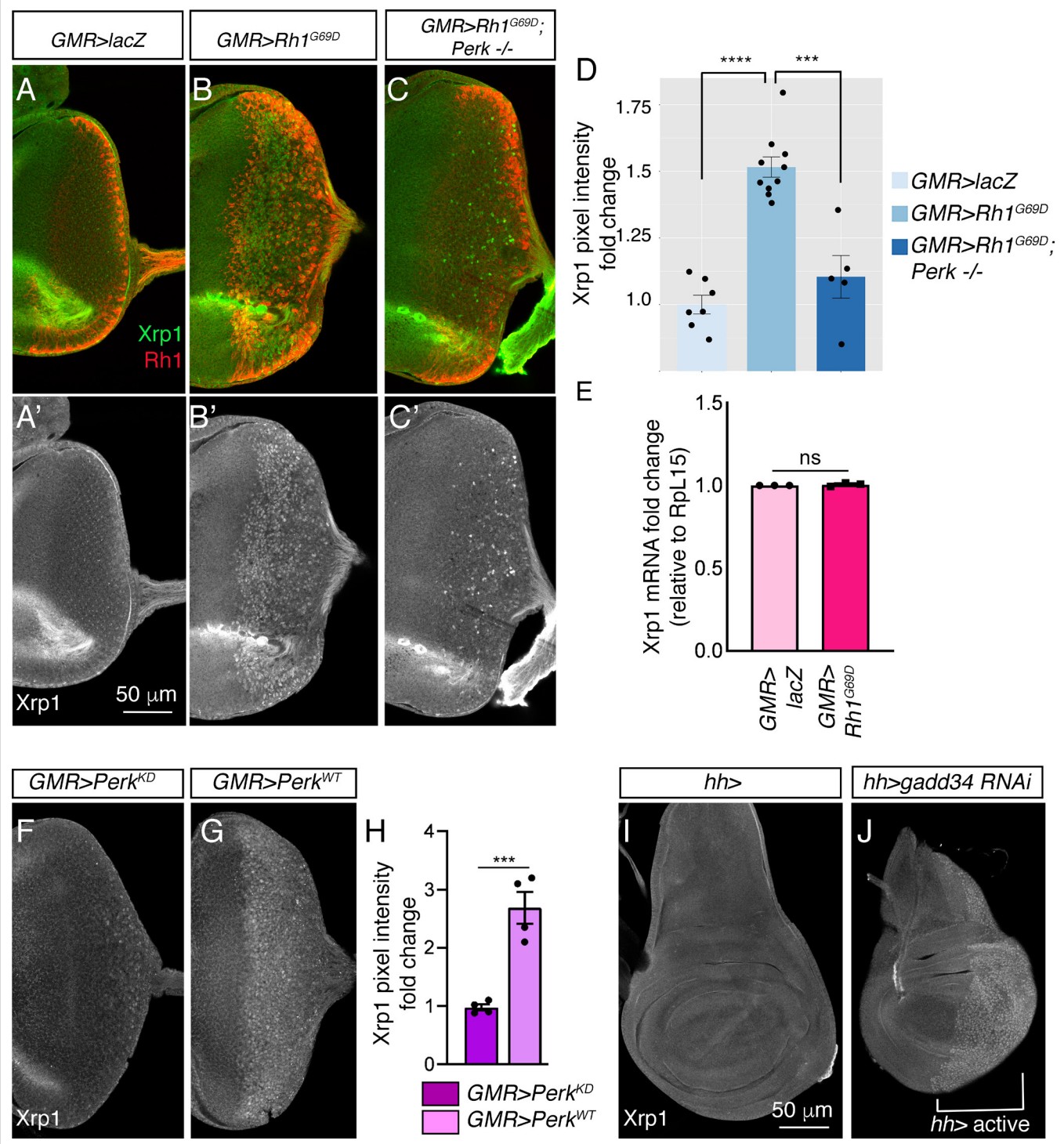

**Figure 6.** Xrp1 induction is regulated by *Perk* and eIF2α phosphorylation. (**A–C**) Eye discs labeled with anti-Xrp1 (green) and anti-Rh1 (red). (**A'-C'**) Anti-Xrp1 single channels of images in (**A–C**). Genotypes: (**A, A'**) *GMR-Gal4; UAS-lacZ/+*. (**B, B'**) *GMR-Gal4; UAS-Rh1^{G69D}/+*. (**C, C'**) *GMR-Gal4; UAS-Rh1^{G69D}/+; Perk^{e01744}*. The scale bar represents 50 μm. (**D**) Quantification of anti-Xrp1 pixel intensities from eye discs in (**A–D**). (**E**) qRT-PCR analysis of *Xrp1* from *GMR > LacZ* (control) and *GMR > Rh1^{G69D}* eye discs. Xrp1 qRT-PCR results were normalized with that of *Rpl15*. (**F, G**) Anti-Xrp1 immunolabeling (white) in eye imaginal discs. Genotypes: (**F**) *GMR-Gal4; uas-Perk^{KD}*. (**G**) *GMR-Gal4; uas-Perk^{WT}*. (**H**) Quantification of anti-Xrp1 pixel intensities in (**F, G**). (**I, J**) Anti-Xrp1 immunolabeling in wing discs. Genotypes: (**I**) *hh-Gal4/+*. (**J**) *uas-gadd34 RNAi/+; hh-Gal4*. The white bracket indicates the posterior compartment where *hh-Gal4* drives the transgene expression. In all graphs, two tailed t tests were used to assess statistical significance. *** = p < 0.0005, **** = p < 0.00005 and ns = non-significant, respectively.

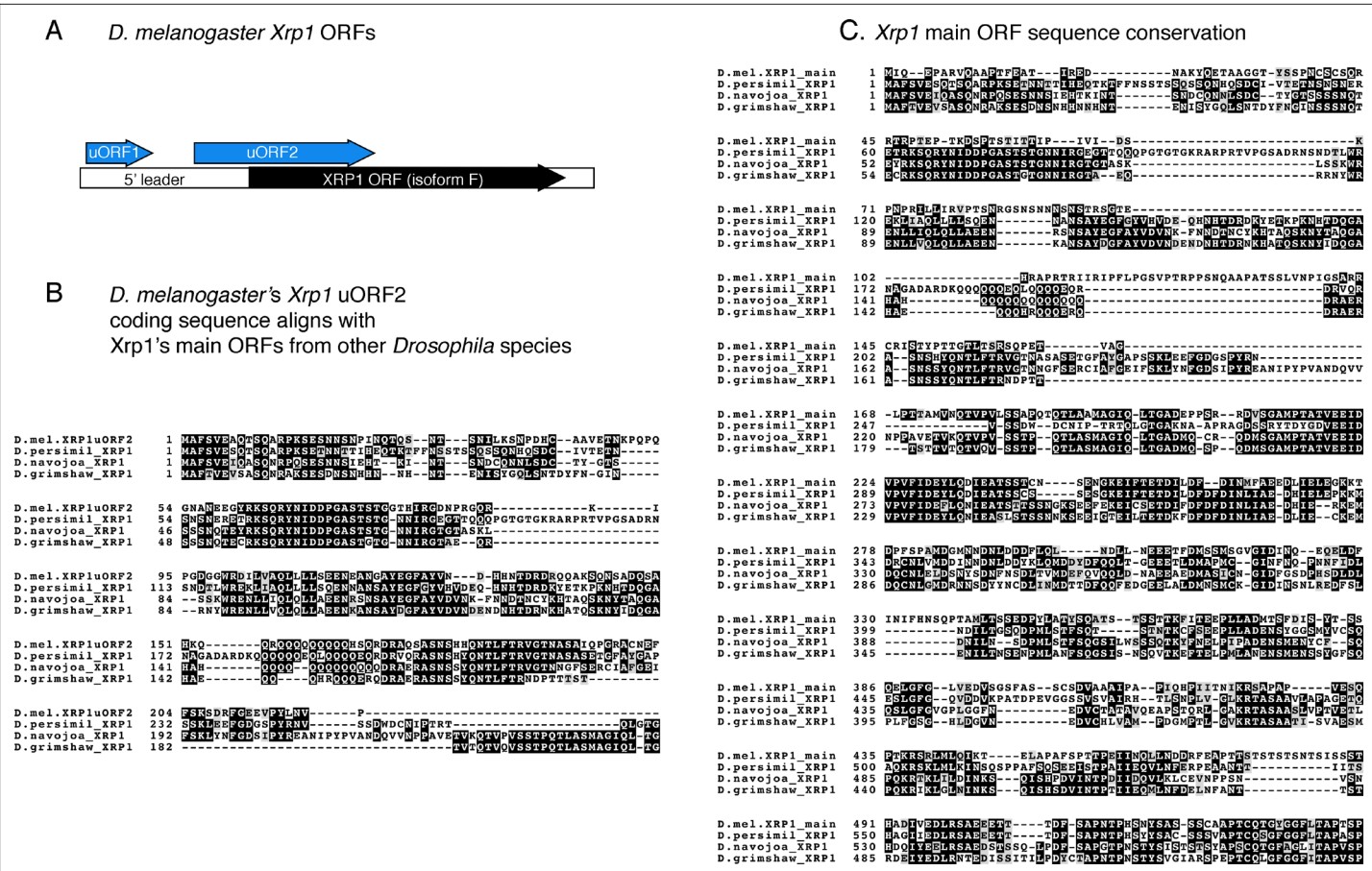

**Figure 7.** Predicted upstream Open Reading Frames (ORFs) in *Xrp1*. (**A**) A schematic diagram of the predicted ORFs in *D. melanogaster Xrp1*. uORF1 and uORF2 in the 5' leader have initiation start codons with Kozak sequences. uORF1 encodes a peptide of 124 amino acid residues. uORF2 encodes a peptide of 288 amino acid residues. uORF2 overlaps with the main Xrp1 ORF, but is in a different reading frame. (**B**) Amino acid sequence alignment between *D. melanogaster Xrp1* uORF2, and the *Xrp1* main ORFs from the species, *D. persimili, D. navojoa, D. grimshawi*. BoxShade was used for visualizing alignments, with those in black indicating sequence identity. The first 219 residues of the *D. melanogaster Xrp1* uORF2 show high-sequence conservation with the main *Xrp1* ORFs from other species. (**C**) Amino acid sequence alignment between the main ORFs of *D. melanogaster* and other *Drosophila* species. *D. melanogaster*'s *Xrp1* main ORF shows high-sequence conservation with other *Drosophila* species beginning from the 140th amino acid residue.

(Percent identity = 50.44%), *D. grimshawi* (Percent identity = 49.48%), and *D. busckii* (Percent identity = 38.69%). The homologous sequences in these other species were part of their Xrp1 main ORF N-terminal regions (*Figure 7B*). The C-terminal regions of these main ORFs all encoded the AT-hook and bZIP DNA-binding domains homologous to that in the *D.* melanogaster's Xrp1 main ORF (*Figure 7C*). The phylogenetic conservation of *D. melanogaster* uORF2 at the peptide level supports the idea that this is a functional uORF that has been under selective pressure during evolution. The similarities in the arrangements of the *D. melanogaster* Xrp1 and ATF4 5' leaders, together with the observation that Xrp1 is induced in response to eIF2α phosphorylation, suggest that Xrp1 and ATF4 share similar mechanisms for their translational induction in response to stress.

## Xrp1 binding sites within the *gstD* enhancer are essential for *gstD-GFP* induction

To determine if Xrp1 regulates *gstD*s directly, we examined the *gstD* 2.7 kb enhancer for putative Xrp1 and ATF4 binding sites (*Figure 8A*). To do so, we used the Xrp1 position frequency matrix derived from a previous Xrp1 ChIP-seq analysis (*Baillon et al., 2018*) and a publicly available ATF4 position frequency matrix (see Materials and methods). Our binding score analysis predicted three putative binding sites each for Xrp1 and ATF4 in the enhancer (*Figure 8B and C*). Two of the three highest

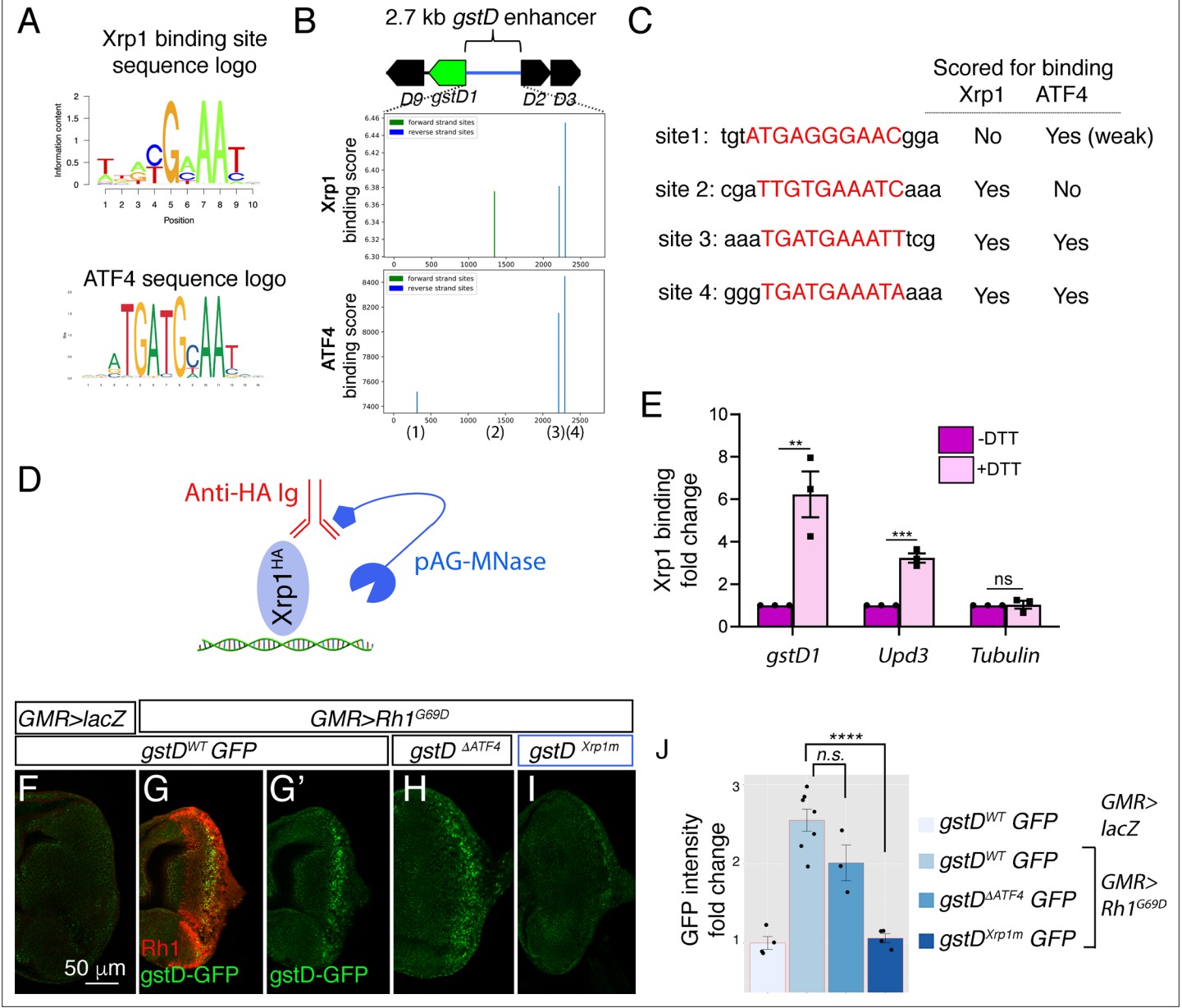

**Figure 8.** Xrp1 binding sites in the *gstD* enhancer are required for *Rh1^G69D* -induced *gstD-GFP* expression. (**A**) Sequence logos representing Xrp1 and ATF4 position frequency matrices. (**B**) Binding score analysis for Xrp1 and ATF4 sites in the *gstD* 2.7 kb enhancer region. Those that score high in the forward strand are depicted in green, whereas those in the reverse strand are shown in blue. The four putative binding sites are numbered below the graphs. (**C**) Putative binding site sequences. The site numbers match those whose positions are indicated in (**B**). (**D**) A schematic diagram of the CUT&RUN approach to assess Xrp1 binding to target DNA. When HA-tagged Xrp1 (light blue) binds to a specific DNA locus, that DNA can be recovered by adding anti-HA antibody (red) and pAG-MNase (dark blue) that cleaves adjacent DNA prior to immunoprecipitation. (**E**) Putative Xrp1 target gene enrichment after Xrp1^HA protein pull down as assessed through q-PCR (using CUT&RUN). The y axis shows fold change increases in target gene DNA recovery in response to DTT treatment (normalized to q-PCR values from controls without DTT treatment). (F - J) Representative eye discs either expressing control *lacZ* (**F**) or those expressing *GMR> Rh1^G69D* (**G–I**), with variants of the gstD-GFP reporter in green: *gstD WT-GFP* (**F, G**) *gstD ^ΔATF4-GFP* (**H**) and *gstD ^Xrp1m-GFP* (**I**). (**G**) shows a merged image with anti-Rh1 (red) and the *gstD-GFP* reporter (green). (**G', H, I**) are gstD-GFP single channel images. (**J**) Quantification of gstD-GFP intensity fold change. Two tailed t tests were used to assess statistical significance. ** = $p < 0.005$, *** = $p < 0.0005$, **** = $p < 0.00001$, n.s. = not significant.

scoring Xrp1 binding sites overlapped with two of the ATF4 sites, while one was predicted to be a unique Xrp1 site.

To test if Xrp1 binds to this locus, we used the CUT&RUN approach (*Skene et al., 2018*) to pull down HA-tagged Xrp1 proteins from *Drosophila* larval tissues to examine if putative Xrp1 target gene

DNAs co-purify (*Figure 8D*, see Materials and methods). We specifically employed an *Xrp1*[HA] transgenic fly line that has an HA-tagged transgene inserted in the endogenous *Xrp1* locus (*Blanco et al., 2020*). Larval tissues incubated with DTT gave strong q-RT-PCR signals for *gstD* regulatory DNA, while control tissues without DTT treatment had minimal DNA recovered. *Upd3* was previously reported as an Xrp1 target gene, and as expected, increased *Upd3* DNA was recovered in response to DTT treatment. Tubulin DNA was used as a negative control, which had minimal DNA recovery even after DTT treatment (*Figure 8E*). These results support the idea that Xrp1 binds to the *gstD* locus in larval cells under ER stress.

We then reconstructed new *gstD-GFP* reporters with (*gstD*[WT]-*GFP*) or without (*gstD*[ΔATF4]-*GFP*) the predicted ATF4-binding sites. To generate an Xrp1 binding site-deficient *gstD-GFP* reporter (*gstD*[Xrp1m]-*GFP*), we introduced mutations in the remaining Xrp1 binding site within the *gstD*[ΔATF4]-*GFP* reporter. To control for genetic backgrounds, we targeted these reporters to a specific attP landing site to generate transgenic flies. While *GMR > Rh1*[G69D] effectively induced both the wild type and *gstD*[ΔATF4]-*GFP* reporters, *gstD*[Xrp1m]-*GFP* was not induced under otherwise identical conditions (*Figure 8F–J*). Together, these results indicate that *gstD* genes are direct targets of Xrp1 in the context of ER stress.

## Discussion

Here, we report that ER stress activates a previously unrecognized UPR axis mediated by PERK and Xrp1. Specifically, we showed that *gstD* family genes are among the most highly induced UPR targets in *Drosophila*, and that such induction requires *Perk*, one of the three established ER stress sensors in metazoans. Surprisingly, the induction of *gstD* genes in this context did not require *crc*, the ATF4 ortholog. Instead, we found that a poorly characterized transcription factor *Xrp1* is induced downstream of *Perk* to promote the expression of *gstDs* and other antioxidant genes.

Our findings are surprising given that ATF4 is considered a major effector of PERK-mediated transcription response (*Karagöz et al., 2019*; *Walter and Ron, 2011*). ATF4 was the first PERK downstream transcription factor to be identified in part based on the similarity of its regulatory mechanisms with that of yeast GCN4 (*Dever et al., 1992*; *Harding et al., 2000*). But more recent studies have shown there could be parallel effectors downstream of PERK activation (*Andreev et al., 2015*; *Baird et al., 2014*; *Palam et al., 2011*; *Zhou et al., 2008*). The functional significance of these alternative factors had remained poorly understood. Our study here has led us to conclude that an ATF4-independent branch of PERK signaling is required for the expression of the most highly induced UPR target in *Drosophila*.

As a potential mediator of this ATF4-independent PERK signaling, we first considered *cncC* as a prime candidate for a few reasons: *cncC* is an established regulator of *gstD-GFP* induction (*Sykiotis and Bohmann, 2008*), and previous studies had reported that Nrf2 is activated by PERK in cultured mammalian cells and in zebrafish (*Cullinan et al., 2003*; *Cullinan and Diehl, 2004*; *Mukaigasa et al., 2018*). However, our results reported here do not support the simple idea that *gstD-GFP* is induced by CncC, which in turn is activated by PERK. Specifically, we found that the loss of *Perk* blocked *gstD-GFP* induction in this experimental setup, but the loss of *cncC* did not. While Nrf2/CncC clearly regulates antioxidant gene expression in response to paraquat, our results indicate that PERK mediates an independent antioxidant response in *Drosophila*.

Our data indicates that this ATF4-independent PERK signaling response requires the AT-hook bZIP transcription factor Xrp1. Several pieces of evidence support the idea that Xrp1 is translationally induced, analogous to the mechanism reported for ATF4 induction. First, our RNA-seq and qRT-PCR results indicate that *Xrp1* transcript levels do not change significantly in *Rh1*[G69D] expressing eye discs. These results argue against the idea that *Xrp1* is induced at the transcriptional level. Second, we find that PERK's kinase domain is required for Xrp1 protein induction. Third, knockdown of *gadd34*, which increases phospho-eIF2α levels downstream of *Perk*, is sufficient to induce Xrp1 protein and *gstD-GFP* expression. Finally, we find that Xrp1's 5′ leader has a uORF that overlaps with the main ORF, similar to what is found in ATF4's regulatory 5′ leader sequence. Moreover, Xrp1's uORF2 encodes a peptide sequence that is phylogenetically conserved in other *Drosophila* species. High-sequence conservation at the peptide level enhances confidence that uORF2 is a peptide coding sequence.

*Xrp1* is known to respond to ionizing radiation, motor neuron-degeneration in a *Drosophila* model for amyotrophic lateral sclerosis (ALS), and in cell competition caused by Minute mutations that cause

haplo-insufficiency of ribosomal protein genes (*Akdemir et al., 2007*; *Baillon et al., 2018*; *Lee et al., 2018*; *Mallik et al., 2018*). Interestingly, two recent studies reported that these Minute cells induce *gstD-GFP*, and also show signs of proteotoxic stress as evidenced by enhanced eIF2α phosphorylation (*Baumgartner et al., 2021*; *Recasens-Alvarez et al., 2021*). Although these studies did not examine the relationships between *Xrp1*, *gstD-GFP* and eIF2α kinases such as *Perk*, our findings make it plausible that the PERK-Xrp1 signaling axis regulates cell competition caused by Minute mutations.

Despite the rising levels of interest in Xrp1 as a stress response factor, the identity of its mammalian equivalent remains unresolved. Xrp1 is well conserved in the Dipteran insects, but neither NCBI Blast searches nor Hidden Markov Model-based analyses identify clear orthologs in other orders (*Akdemir et al., 2007*; *Blanco et al., 2020*; *Mallik et al., 2018*; *Baillon et al., 2018*). Such evolutionary divergence is not unprecedented in UPR signaling: GCN4 is considered a yeast equivalent of ATF4, but they are not the closest homologs in terms of their peptide sequences (*Harding et al., 2000*). Likewise, the yeast equivalent of XBP1 (IRE1 effector, not to be confused with Xrp1 in this study) is Hac1, but there is little sequence conservation between the two genes (*Yoshida et al., 2001*; *Shen et al., 2001*; *Calfon et al., 2002*). Yet, the UPR signaling mechanisms are considered to be conserved due to the shared regulatory mechanisms. Along these lines, mammalian cells may have functional equivalents of Xrp1. We consider among the candidate equivalent factors those with regulatory 5′ leader sequences that respond to eIF2α phosphorylation (*Palam et al., 2011*; *Zhou et al., 2008*; *Andreev et al., 2015*). Based on the emerging roles of *Xrp1* in *Drosophila* models of human diseases, we speculate that those ATF4-independent PERK signaling effectors may play more significant roles in diseases associated with UPR than had been generally assumed.

We note in our study that genes encoding cytoplasmic glutathione S-transferases (GSTs) such as *gstD1* and *gstD9* are among the most prominently induced UPR targets in our eye imaginal disc-based gene expression profiling analysis. Previous studies also reported these as ER stress-inducible genes in *Drosophila* S2 cells (*Malzer et al., 2018*). GSTs are cytoplasmic proteins that participate in the detoxification of harmful, often lipophilic intracellular compounds damaged by ROS. These enzymes catalyze the formation of water-soluble glutathione conjugates that can be more easily eliminated from the cell (*Low et al., 2010*; *Low et al., 2007*; *Sharma et al., 2004*; *Wilce and Parker, 1994*). It is noteworthy that ROS is generated as a byproduct of Ero-1-mediated oxidative protein folding, and such ROS generation increases when mutant proteins undergo repeated futile cycles of protein oxidation (*Gross et al., 2006*; *Tang and Tu, 1994*; *Tu and Weissman, 2004*). Therefore, we speculate that cytoplasmic GSTs evolved as UPR targets as they have the ability to detoxify lipid peroxides or oxidized ER proteins that increase in response to ER stress.

In conclusion, our findings support the idea that an ATF4-independent branch of PERK signaling mediates the expression of the most highly induced UPR targets in eye disc cells. This axis of the UPR requires *Xrp1*, a gene that had not previously been associated with ER stress response. The identification of this new axis of UPR signaling may pave the way for a better mechanistic understanding of various physiological and pathological processes associated with abnormal UPR signaling in metazoans.

# Materials and methods
## Fly stocks and husbandry

We reared flies at ambient temperature on a standard cornmeal-agar diet supplemented with molasses, and carried out crosses at 25 °C. We performed all gene overexpression experiments using the Gal4/UAS binary expression system (*Brand and Perrimon, 1993*). We used the following fly stocks: *w1118*, *gstD-GFP* (*Sykiotis and Bohmann, 2008*), *GMR-Gal4* (*Freeman, 1996*), *Rh1-Gal4* (*Pichaud and Desplan, 2001*), *hh-Gal4* (*Calleja et al., 1996*), *UAS-EGFP::Msp300KASH* (*Hall et al., 2017*; *Ma and Weake, 2014*; RRID: BDSC_92584), *UAS-Rh1G69D* (*Ryoo et al., 2007*), *UAS-PerkWT* (RRID: BDSC_76248), *UAS-PerkKD* (*Malzer et al., 2010*; RRID: BDSC_76249), *UAS-cyto-roGFP-Orp1* (*Albrecht et al., 2011*; RRID: BDSC_67670), *ey-FLP* (RRID: BDSC_5576), *FRT82B,arm-lacZ* (RRID: BDSC_7369), *FRT82B,p(w+)[90E]* (RRID: BDSC_2050), *UAS-lacZ* (RRID: BDSC_1776), *FRT82B,cncK6/TM6B* (*Veraksa et al., 2000*), *FRT82B, cncvl110* (*Mohler et al., 1995*), *FRT82B,Ire1f02170/TM6B* (*Ryoo et al., 2013*; RRID: BDSC_18520), *FRT82B,Perke01744* (*Wang et al., 2015*; RRID: BDSC_85557), *FRT82B, Xrp1M2-73* (*Lee et al., 2018*; RRID: BDSC_81270), *Xrp1HA* (*Blanco et al., 2020*), *Atf6LL07432* (Kyoto DGGR,

#142049), *crc¹* (*Hewes et al., 2000*; RRID: DGGR_105823), *Thor^k13517* (*Thor-lacZ*)(*Bernal and Kimbrell, 2000*; RRID: BDSC_9558), *uas-crc^leaderless* (*Vasudevan et al., 2020*), *uas-gadd34 RNAi* (VDRC #107545). The *gadd34* RNAi line was expressed together with *uas-dicer2* on the 3rd chromosome to enhance knockdown efficiency.

## Identification of ATF4 and Xrp1 binding sites in the *gstD1/gstD2* intergenic enhancer

We modified an existing Python code that that uses TBP position frequency matrix to predict TATA boxes (*Stevens and Boucher, 2015*) to calculate transcription factor binding scores. We deposited this modified Python code in Github (https://github.com/finnroach/transcription-factor-binding; copy archived at swh:1:rev:32ecddcc5cd7c8a22b2bb3db073162fc62d79447; *Finnegan, 2021*).

To identify putative ATF4 binding sites in the *gstD1* upstream enhancer, we utilized the position frequency matrix (PFM) information of human ATF4 available on JASPAR (jaspar.genereg.net, Matrix ID:MA0833.1). The code outputs only binding sites with scores that are greater than 76 % of the optimal ATF4 binding sequence. This cutoff both picks up two known ATF4 binding sites within the *Drosophila Thor* intron sequence (*Kang et al., 2017*) and allows leeway to find slightly lower affinity binding sites. Inputting the *gstD1* enhancer sequence into this code revealed three unique putative ATF4 binding half-sites with the following sequences: cgttccctcatac (77 % of optimal binding score), aatttcatcattt (83 % of optimal binding score), and tatttcatcaccc (86 % of optimal binding score).

To score putative Xrp1 binding sites, we used information available from a previous Xrp1 ChIP-seq study. The sequence logo of Xrp1 position frequency matrix was reported previously (*Baillon et al., 2018*). The Xrp1 position frequency matrix itself is available from the link (https://www.biorxiv.org/content/10.1101/467894v1). We show in *Figure 8* the outputs of sites with scores greater than 95 % of the optimal Xrp1 binding sequence.

## Generation of transgenic lines: *gstD^WT-GFP*, *gstD^ΔATF4-GFP*, and *gstD^Xrp1m-GFP*

To construct the *gstD^WT-GFP* reporter, we amplified the entire genomic region of *gstD-GFP* flies containing the *gstD-GFP* reporter, then cloned the amplicon into BglII/NotI-digested pattB using InFusion cloning (ClonTech). To make *gstD^ΔATF4-GFP*, we used commercial gene synthesis (General BioSystems, Inc) to reconstruct the entire reporter region previously amplified by PCR for the creation of *gstD^WT-GFP*, but with the predicted ATF4-binding sites deleted. The three deleted sequences are (in the order of from that closest to *gstD2* towards the *gstD1* coding sequence): ggtgatgaaata, aatgatgaaatt, atgagggaa.

To generate the *gstD^Xrp1m-GFP* reporter, we performed directed mutagenesis on the *gstD^ΔATF4-GFP* to mutate the remaining Xrp1 binding site. Specifically, the sequence ttgtgaaatc was mutated to ttcccgggtc. The wild type and mutant *gstD-GFP* DNA were then subcloned into the pattB vector, and sent to BestGene.Inc (Chino Hills, CA) for embryo injection. Standard phiC31 integrase-mediated germline recombination approach (*Groth et al., 2004*) was used to target the plasmids to an attP landing site at cytological position 51 C (Bloomington Stock Center, #24482).

## Affinity-based isolation of EGFP-labeled nuclear membranes and RNA extraction

We followed published protocols to isolate GMR > EGFP::Msp-300^KASH-positive nuclei (*Hall et al., 2018*; *Ma and Weake, 2014*). We dissected eye-antennal discs from approximately 100 larvae per sample in phosphate buffered saline (PBS), pH 7.4 with 0.1 % Tween-20 (Sigma-Aldrich, cat. #P7949). We then washed dissected discs in ice-cold nuclear isolation buffer (10 mM HEPES-KOH, pH 7.5; 2.5 mM MgCl$_2$; 10 mM KCl), then lysed cells in 1 mL of ice-cold nuclear isolation buffer in a 2 mL Dounce homogenizer (VWR, cat. #62400–595). We next filtered the homogenate through a 40 µm Flowmi cell strainer (WVR, cat. #BAH136800040), after which we incubated a 20 µL pre-isolation aliquot of the filtrate with 0.1 µg/mL 4′,6-diamidino-2-phenylindole (DAPI) (Millipore-Sigma, cat. #D9542) and mounted on a slide for imaging (Fisher, cat. #12-550-433). We incubated the remaining filtrate with anti-EGFP-coupled protein G Dynabeads (Invitrogen, cat. #10,003D) for 1 hr at 4 °C with gentle end-over-end rotation. Next, we collected the beads using a magnetic microcentrifuge tube holder (Sigma, cat. #Z740155) and washed the collected beads with wash buffer (PBS, pH 7.4; 2.5 mM

MgCl$_2$), then resuspended the beads in a final volume of 150 μL of wash buffer. We incubated an aliquot of the post-isolation sample with 0.1 μg/mL DAPI and mounted the sample on a slide for imaging. Finally, we suspended the post-isolation nuclei in 1 mL of Trizol reagent (Life Technologies, cat. #15596018) for RNA extraction following standard procedures. Prior to RNA precipitation with isopropanol, we added 0.3 M sodium acetate and glycogen (Invitrogen, cat. #AM9510) to facilitate visualization of the RNA pellet. We then suspended the pellet in RNAse-free water and purified it using a Qiagen RNeasy MinElute cleanup kit (Qiagen, cat. #74204) following standard protocols.

## Preparation of cDNA libraries, RNA-Seq, and data processing

The NYU Genome Technology Center performed library preparation and RNA sequencing. We quantified RNA on an Agilent 2100 BioAnalyzer (Agilent, cat. #G2939BA). For cDNA library preparation and ribodepletion, we utilized a custom *Drosophila* Nugen Ovation Trio low-input library preparation kit (Tecan Genomics), using approximately 1.5 ng total RNA per sample. For sequencing, we performed paired-end 50 bp sequencing of samples on an Illumina NovaSeq 6000 platform (Illumina, cat. #20012850) using half of a 100 cycle SP flow cell (Illumina, cat. #20027464). We used the bcl2fastq2 Conversion software (v2.20) to convert per-cycle BCL base call files outputted by the sequencing instrument (RTA v3.4.4) into the fastq format in order to generate per-read per-sample fastq files. For subsequent data processing steps, we used the Seq-N-Slide automated workflow developed by Igor Dolgalev (https://github.com/igordot/sns; *Dolgalev, 2021*). For read mapping, we used the alignment program STAR (v2.6.1d) (*Dobin et al., 2013*; *Dobin and Gingeras, 2015*) to map reads of each sample to the *Drosophila melanogaster* reference genome dm6, and for quality control we used the application Fastq Screen (v0.13.0) (*Wingett and Andrews, 2018*) to check for contaminating sequences. We employed featureCounts (Subread package v1.6.3) (*Liao et al., 2014*) to generate matrices of read counts for annotated genomic features. For differential gene statistical comparisons between groups of samples contrasted by genotype, we used the DESeq2 package (R v3.6.1) (*Love et al., 2014*) in the R statistical programming environment. For filtering of stably induced genes, we calculated the coefficient of variation (CV) of FPKM-normalized counts for all genes across all three *GMR > Rh1$^{G69D}$* samples, and excluded genes with a CV greater than 0.3 from downstream analyses (*Liang et al., 2020*).

## Antibodies, and immunofluorescence and confocal microscopy

To generate a guinea pig anti-Xrp1 antibody, we expressed the Xrp1 long isoform (isoform F) cloned in pSV272 (*Francis et al., 2016*) in BL21 pLys *E. coli* strain by IPTG induction. After cells lysis and sonication in pre-chilled binding buffer (20 mM Tris pH 8.0, 0.5 M NaCl, 5 mM imidazole), we centrifuged the lysate and washed the insoluble pellet containing His-tag fused recombinant Xrp1 in wash buffer (20 mM Tris pH 8.0, 0.5 M NaCl, 5 mM imidazole), and then solubilized the pellet in denaturing buffer (8 M Urea with 20 mM Tris pH 8.0 and 100 mM NaCl). We ran the solubilized fraction containing His-Xrp1 through a Ni$^{2+}$-NTA column and washed the column with reducing concentrations of Urea in the denaturing buffer for purification and re-folding. We eluted His-Xrp1 from the column with Tris buffer containing 400 mM imidazole, and sent the purified protein for custom polyclonal antibody production (Covance Inc). The crude antiserum was affinity purified and used at 1:10 dilution for immunolabeling.

We immunoprecipitated EGFP-labeled imaginal disc nuclei with mouse anti-EGFP (Roche, cat. #11814460001). For immunofluorescence, we used the following primary and secondary antibodies at the indicated dilutions: rabbit anti-β-Gal (1:500, MP Biomedicals, cat. #55976), Rabbit anti-GFP (1:500 Invitrogen #A6455), chicken anti-EGFP (1:500, Aves lab, cat. #GFP-1020), mouse anti-Rh1 (1:500, Developmental Studies Hybridoma Bank (DSHB), 4C5 concentrate), rat anti-Elav (1:50, DSHB, 7E8A10 concentrate), mouse anti-Wingless (1:25, DSHB, 4D4), mouse-anti-HA (1:500, Cell Signaling, 2,367 S), Alexa Fluor 647 goat anti-mouse (1:500, Life Technologies, cat. #A-21235), Alexa Fluor 488 goat anti-chicken (1:500, Life Technologies, cat. #A-11039), Alexa Fluor 546 goat anti-rabbit (1:500, Life Technologies, cat. #A-11035).

We followed standard protocols for whole-mount immunofluorescence. We fixed discs in 1 X PBS with 0.2 % Triton X-100 (Millipore-Sigma, cat. #T8787) (PBTx 0.2%) and 4 % paraformaldehyde (Alfa Aesar, cat. #43368) for 20 min at ambient temperature with gentle rocking. Next, following three short rinses with PBTx 0.2%, we incubated discs with primary antibodies in PBTx 0.2 % for 1 hr at

ambient temperature with gentle rocking. Following primary antibody incubation, we washed discs for 3*10 min with PBTx 0.2%, then covered from light and incubated with secondary antibody in PBTx 0.2 % for 1 hr at ambient temperature with gentle rocking. Following secondary antibody incubation, we washed discs 3*10 min with PBTx 0.2%, then mounted in 50 % glycerol (Millipore-Sigma, cat. #G5516) with 0.1 µg/mL DAPI. For all confocal micrographs, we captured images on a Zeiss LSM 700 confocal microscope (Carl Zeiss). For image acquisition, we scanned all imaginal discs under a 40 X water objective, and all isolated nuclei under a 100 X oil objective.

## S2 cell culture, RNAi, and drug treatments

We cultured *Drosophila* S2 cells in ambient conditions in S2 cell medium (Fisher, cat. #21720024) supplemented with 1 % penicillin/streptomycin (Life Technologies, cat. #15-140-122) and 10 % heat-inactivated FBS (Fisher, cat. #10082147). For maintenance of cell lines, we passaged cells every 3–4 days and utilized them for experiments between passages 6 and 15.

We performed RNAi using a modified dsRNA bathing protocol (*Ryoo et al., 2007*). We generated dsRNA against *cncC*, *Perk, Xrp1* and *crc* mRNAs following an established T7 in vitro transcription protocol (ThermoFisher, cat. #AM1334) (*Armknecht et al., 2005*). On day 0, we added approximately 20 µg of the indicated dsRNA to 1 mL containing $10^6$ cells in serum-free S2 cell medium in a six-well dish. After 30 min of incubation at ambient temperature, we added 3 mL of S2 cell medium with serum and incubated the cells at ambient temperature. On day 3, we harvested the cells, resuspended them at $1*10^6$ cells/mL in serum-free S2 cell medium, and subjected them to another round of dsRNA bathing as described above. On day 6, we re-plated cells at a concentration of $\sim 2*10^6$ cells/mL for drug treatments. For drug treatments, we exposed cells to either 20 mM paraquat (Sigma, cat. #856177) or 10 µg/mL tunicamycin (Fisher, cat. #351610) for the indicated times, then harvested and resuspended the treated cells in 500 µL Trizol reagent. Following RNA isolation, we then treated the samples with Turbo DNAse (Invitrogen, cat. #AM1907) for 25 min at 37 °C to remove traces of contaminating genomic DNA. We utilized the following oligos to generate T7 double-stranded RNA:

    EGFP-F: TAATACGACTCACTATAGGGAACAGCCACAACGTCTATATC
    EGFP-R: TAATACGACTCACTATAGGGTTGGACAAACCACAACTAGAA
    PERK-F: TAATACGACTCACTATAGGGTGGCACAAGGAGGGGAAC
    PERK-R: TAATACGACTCACTATAGGGGCACCACTGGACCTAGTAAA
    CncC-F: TAATACGACTCACTATAGGGGGGCTGCAAGCTTCCG
    CncC-R: TAATACGACTCACTATAGGGGCGGTGCTGAGGGGTG
    ATF4-F: TAATACGACTCACTATAGGGGCGGTGTAGAGGATCGAAAG
    ATF4-R: TAATACGACTCACTATAGGGCACTGTCCGATTTGCAGAAA
    Xrp1-F: TAATACGACTCACTATAGGGAGGACGAAGAGGAGACTACCACCG
    Xrp1-R: TAATACGACTCACTATAGGGAGGAGTAAGTGCTCTTCTGCCGCT

## Molecular biology and qRT-PCR

We amplified the gstD-GFP reporter from transgenic fly genomic DNA for InFusion cloning into BglII/NotI-digested pattB using the following oligos: gstD-GFP-F: attcgttaacagatcattgcaactggttgttaacct gstD-GFP-R: cctcgagccgcggcccgccttaagatacattgatgagt.

For qRT-PCRs, we reverse transcribed approximately 500 ng of total RNA from S2 cells in a 20 µL reaction with random primers using Maxima H reverse transcriptase (Thermo Scientific, cat. #EP0752) following manufacturer protocols. From this, we used 1 µL of cDNA per well for qPCR on a BioRad CFX96 Touch Real-Time PCR Detection System (BioRad, cat. #1855196) using Power SYBR Green PCR master mix (Life Technologies, cat. #4367659). We determined cycle threshold ($C_t$) values with BioRad CFX Manager software. For mRNA fold change calculations, we used the $\Delta\Delta C_t$ method, normalized to the housekeeping gene *RpL15* (*Livak and Schmittgen, 2001*). We used the following oligos for qRT-PCR:

    RpL15-F: AGGATGCACTTATGGCAAGC
    RpL15-R: GCGCAATCCAATACGAGTTC
    Thor-F: TAAGATGTCCGCTTCACCCA
    Thor-R: CGTAGATAAGTTTGGTGCCTCC gstD2-F: CCGTCTATCTGGTGGAGAAGTA gstD2-R: GAGTTCCCATGTCGAAGTACAG
    EGFP::Msp-300^KASH-F: GAGGGATACGTGCAGGAGAG

EGFP::Msp-300[KASH]-R: GATCCTGTTGACGAGGGTGT
Gstd1-F: CATCGCGAGTTTCACAACAG
Gstd1-R: GTTGAGCAGCTTCTTGTTCAG
Gstd9-F: TTGCCGTTCCATCCTGATGAC
Gstd9-R: GCTTAAGATGCTCGCCGGCAT
Gstd10-F: TGCCGCTCTGTTCTGATG
Gstd10-R: CTAGCTCGGGTGTTGATAAT
SOD2-F: TCGCAAACTGCAAGCCTG
SOD2-R: CATGATCTCCCGGCAGAT
JAFRAC1-F: ATCATTGCGTTCTCGGAG
JAFRAC1-R: GCGTGTTGATCCAGGCCAA

## Quantification of gstD-GFP pixel intensity changes

We used ImageJ (https://imagej.nih.gov/ij/index.html) to calculate average pixel intensities of gstD-GFP signals from eye disc microscopic images. We specifically measured pixel intensities of the *GMR-Gal4* expressing region in posterior eye discs. To calculate fold change of the gstD-GFP signal, we divided the average pixel intensities of interest with that from control discs. Statistical significance was assessed through two tailed t tests unless otherwise stated. We plotted graphs using the ggplot2 package in the R programming environment.

## Sample preparation for CUT and RUN

Approximately, 100 tissues including eye, wing, leg discs and brains for each experimental condition were isolated from third instar wandering larvae (Genotype: *hspflp; FRT42/Cyo;Xrp1[HA-1]/TM6B*). The isolated tissues were soaked in 2 mM DTT for 6 hr in S2 media at RT in shaking condition. After incubation, samples were centrifuged at 1200 rpm for 5 min at 4° and 1 X trypsin was added to the pellet to isolate single-cell suspension. Trypsin digestion was continued for 4 hr at RT in a nutator. After digestion, the reaction mixture was centrifuged at 1200 rpm for 5 min and the isolated cells were re-suspended in cold wash buffer (2 % FBS in 1 X DPBS without Ca/Mg and EDTA). The cells were kept on ice from this step. The cell suspension was filtered using a 40 µm cell strainer. Isolated cells were counted in haemocytometer and approximately 250,000 cells/sample were used for further processing using CUTANA ChIC/ CUT & RUN kit (Epicypher, 14–1048) according to the manufacture's protocol. For IP reaction of HA-tagged Xrp1, ChIP grade anti-HA antibody (Abcam, ab9110) was used. As a control isotype, ChIP grade control IgG was used (Abcam, ab171870).

## Validation of Xrp1 binding to gstD locus by qPCR assay from CUT and RUN sample

A total of 500 picogram of eluted DNA was used as a template for qPCR analysis in a 20 µL reaction with the following primers on a BioRad CFX96 Touch Real-Time PCR Detection System (BioRad, cat. #1855196) using Power SYBR Green PCR master mix (Life Technologies, cat. #4367659). We determined cycle threshold ($C_t$) values with BioRad CFX Manager software. For relative fold change in binding event calculations, $\Delta C_t$ value was evaluated in comparison to the control IgG for each condition, while $\Delta\Delta C_t$ was generated by normalizing to unstressed condition (Without DTT). Here, UPD3 was used as a known binding partner of Xrp1 (*Baillon et al., 2018*) and Tubulin was used as an Xrp1 unbound chromatin control. We used the following oligos for qRT-PCR:

GstD1 F: TTGCGCTCTTAACGTCGAGAAC
GstD1 R: CAGCTGGATTTCGGCATTATGT
UPD3 F: TCGATACTTCTGAACCCGC
UPD3 R: CCTAAACCCATCACACCT
Tubulin F: AGCTCGATAACTCCGCATTGGC
Tubulin R: AGAGCGAGACGGCCGAT

## Acknowledgements

We thank Nicholas Baker, Claude Desplan, Michael Garabedian, Moses Chao, Erika Bach and Jessica Treisman for helpful comments. We also thank Heinrich Jasper, Nicholas Baker, Konrad Basler, Donald

Rio, Thomas Hurd, Temesgen Fufa, Robert Hufnagel, Arjita Sarkar and Vikki Weake for fly strains, plasmids and technical advice. This project was supported by NIH R01 EY020866 and GM125954 to HDR, T32 HD007520 and T32 GM136573 training grants support for BB, K99 EY029013 to DV, and the Cancer Center Support Grant P30 CA061087 at the Perlmutter Cancer Center to the Genome Technology Center.

## Additional information

### Funding

| Funder | Grant reference number | Author |
| --- | --- | --- |
| National Eye Institute | R01 EY020866 | Hyung Don Ryoo |
| National Institute of General Medical Sciences | R01 GM125954 | Hyung Don Ryoo |
| National Institute of General Medical Sciences | T32 GM136573 | Brian Brown |
| Eunice Kennedy Shriver National Institute of Child Health and Human Development | T32 HD007520 | Brian Brown |
| National Eye Institute | K99 EY029013 | Deepika Vasudevan |
| NYU School of Medicine | P30 CA061087 | Deepika Vasudevan |

The funders had no role in study design, data collection and interpretation, or the decision to submit the work for publication.

### Author contributions

Brian Brown, Investigation, Writing - original draft; Sahana Mitra, Investigation, Writing - original draft, Writing – review and editing; Finnegan D Roach, Software, Visualization, Writing – review and editing; Deepika Vasudevan, Investigation, Writing – review and editing; Hyung Don Ryoo, Conceptualization, Funding acquisition, Investigation, Project administration, Software, Supervision, Writing - original draft, Writing – review and editing

### Author ORCIDs

Brian Brown http://orcid.org/0000-0001-9826-4052
Sahana Mitra http://orcid.org/0000-0001-8339-1307
Finnegan D Roach http://orcid.org/0000-0002-4214-2002
Hyung Don Ryoo http://orcid.org/0000-0002-1046-535X

### Decision letter and Author response

Decision letter https://doi.org/10.7554/eLife.74047.sa1
Author response https://doi.org/10.7554/eLife.74047.sa2

## Additional files

### Supplementary files

- Transparent reporting form
- Supplementary file 1. Supplementary table 1.
- Source data 1. Supplementary Source Data.

### Data availability

Sequencing data have been deposited in GEO under the accession code GSE150058. Source Data files have been provided for Figures 2-6 and 8.

The following dataset was generated:

| Author(s) | Year | Dataset title | Dataset URL | Database and Identifier |
|---|---|---|---|---|
| Brown BS, Ryoo H | 2021 | Transcriptional changes associated with endoplasmic reticulum stress in the eye imaginal disc of *Drosophila melanogaster* | https://www.ncbi.nlm.nih.gov/geo/query/acc.cgi?acc=GSE150058 | NCBI Gene Expression Omnibus, GSE150058 |

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
