## [Decision Letter]

[Editors' note: this paper was reviewed by Review Commons.]

**Acceptance summary:**

This work describes a new effector of the unfolded protein response, which allows cells to maintain homeostasis in the endoplasmic reticulum. ER stress activates PERK, which broadly attenuates translation while also specifically increasing translation of the transcription factor Atf4. The authors convincingly demonstrate that a second transcription factor, Xrp1, is also upregulated by PERK and regulates antioxidant-related gene expression in flies. This opens up a new mechanism through which organisms can maintain proteostasis and respond effectively to stress.

---

## [Author Response]

We thank the Review Commons reviewers for the overall enthusiastic assessments regarding our manuscript.

Specifically, Reviewer #1 summarized our work as “describing a novel signaling pathway downstream of PERK,” and added that “the conclusion of this paper should be interesting to a broad readership interested in the mechanisms supporting proteostasis.” Reviewer #2 commented that “overall, the experiments were well designed, executed and included detailed analysis,” and that our manuscript is the “first demonstration of a PERK target that is independent of ATF4.” Reviewer #3 also expressed enthusiasm for our manuscript, stating that “by discovering an entirely novel signaling route downstream of PERK that is independent of ATF4, this broadens our understanding of the pathway, making a major contribution to our understanding of the Unfolded Protein Response,” adding that the “data are high quality and clear,” and that the conclusions drawn from the data are well justified.”

At the same time, Reviewer #2 asked about the mechanism by which PERK induces Xrp1. During the past few months, we have generated significant amounts of additional data providing insights into the mechanism of Xrp1 induction by PERK. These data are now included in our revised Figures. We hope you agree that these additions address virtually all points raised by the reviewers.

Below, please find a summary of the changes that we have made in response to reviewer comments.

Reviewer #1:This paper describes a novel signaling pathway downstream of the kinase PERK which mediates transcriptional responses to unfolded protein stress. The presented data show convincingly that the bZIP transcription factor Xrp1 is required for the activation of gene expression to protect cells against the negative effects of excess unfolded proteins in the ER. The target genes that are shown to be upregulated by Xrp1 in a PERKdependent manner include several, such as GstD1, that are known to mediate antioxidant responses as targets of the transcription factor Nrf2, CncC in *Drosophila*.Brown et al. show that neither CncC nor the *Drosophila* homolog of ATF4, which is the best-known transcriptional effector of PERK are involved in mediating the PERK response to over-expression of a misfolded rhodopsin in the larval eye imaginal disc or other forms of ER stress in cultured cells.The data supporting the surprising discovery that Xrp1 is a specific downstream of Perk are well controlled. The conclusion of this paper should be interesting to a broad readership interested in mechanisms supporting proteostasis, which is increasing recognized for its role in organism health and successful aging.

We thank the reviewer for the positive assessment of our work.

Publication should be considered. However, some clarifications in the description of the data and the discussion would improve the manuscript.1. The authors make a point that the induction of the GstD1 transcriptional reporter in response to Rh1G69D expression in the eye disc is predominantly seen in non neuronal cells. Is the same observed when discs are exposed to ER stress by incubation with DTT as shown in Figure S2?

We address this point in our revised Figure S2A-D, where we show that ER stress causing chemicals also induce the gstD-GFP reporter primarily in nonneuronal cells, as demonstrated by co-labeling with Elav.

2. Figure S3: It looks as if the lacZ negative, cnc-/- clones show a stronger GFP response than clones that have a functional cnc gene. Is this the case? If so why? What is the blue channel? Rh1?

The reviewer is correct in that cnc -/- clones show stronger gstD-GFP induction. We now comment on this in the revised manuscript (page 9), by writing that “contrary to our expectations, cncC mutant clones showed enhanced gstD-GFP expression as compared to the neighboring wild type cells (Figure S3A-C). This enhanced gstD-GFP expression in the mutant clones could reflect increased proteostatic stress associated with *cncC* loss. Based on these data, we conclude that *cncC* is not required for gstD-GFP expression in response to ER stress.”

3. Figure 3: Several areas of lacZ positive, presumably PERK expressing cells do not show GFP expression, whereas in control GMR Gal4>Rh1 eye disc GstGFP seems to induced rather ubiquitously. Why is that?

The point we could conclude with confidence here is that the Perk mutant clones abolish gstD-GFP expression, whereas the control cells have residual reporter signals. Regarding the slightly weaker gstD-GFP reporter expression in lacZ positive cells, there could be several different possibilities (the genetic background of the lacZ chromosome, versus a non-autonomous effect, etc.). Because we do not have data to support a specific possibility, we decided to focus on the main point and not speculate on possible non-autonomous effects.

4. Page 10, lines 15 – 17: "To our surprise, loss of crc did not impair gstD-GFP induction in the eye discs (Figure 4B',C') expressing Rh1G69D to similar extents (Figure 4B',C')." This should be rewritten. To similar extents as what? Probably refers to Thor lacZ. But that is not clear as written. Also: does crc over expression activate thor lacZ? That would be a good control.

We now changed the sentence to “Rh1 G69D expression levels were similar in discs with or without crc.” Regarding the following suggestion regarding crc overexpression, we now show in the revised Figure 4I that crc overexpression activates Thor-lacZ expression.

5. Figure 5c: Similar as pointed out above for Figure 3: There are many areas that are not Xrp negative clones, with very few GFP positive cells. Isn't that unexpected? Is there some kind of a non-autonomous effect?

As noted in response to point #3, our existing data does not allow us to conclude with confidence whether Xrp1 exerts a non-autonomous effect.

6. Discussion: Xrp1 has recently been shown to be a critical mediator of the minute phenotype resulting from haplo-insuffiency of ribosomal protein genes. It is required to mediate cell competition during development and may play a role in genome surveillance (Bianco, Cooper and Baker, 2019). While it is not clear if and how these exciting insights relate to the ER stress response function of Xrp1 described here, it would be interesting for the readers to discuss this in the manuscript.

We thank the reviewer for raising this interesting point. Perk is an eIF2a kinase, and two papers came out in late January 2021 indicating that cell competition is associated with proteotoxic stress that causes eIF2a phosphorylation. Based on this, we added the following discussion in page 20 of the revised manuscript:

“Xrp1 is a gene known to respond to ionizing radiation, motor neurondegeneration in a *Drosophila* model for amyotrophic lateral sclerosis (ALS), and in cell competition caused by Minute mutations that cause haplo-insufficiency of ribosomal protein genes (Akdemir, 2007; Lee, 2018 #1106; Baillon, 2018; Mallik, 2018). Interestingly, two recent studies reported that these Minute cells induce *gstD-GFP*, and also show signs of proteotoxic stress as evidenced by enhanced eIF2a phosphorylation (Recasens-Alvarez, 2021; Baumgartner, 2021). Although any possible relationships between *Xrp1*, *gstD-GFP* and eIF2a kinases such as *Perk* were not examined in those studies, our findings make it plausible that the PERK-Xrp1 signaling axis regulates cell competition caused by Minute mutations.”

We would also like to highlight the fact that two other independent studies associating with Xrp1 and ER stress response during cell competition are currently under review. One of those is being considered in *eLife* (manuscript by Nicholas Baker, “The transcription factor Xrp1 orchestrates both reduced translation and cell competition upon defective ribosome assembly or function.”). We could cite that article upon their publication.

Significance:As indicated above, this paper extends our understanding of mechanisms that mediate the cellular responses to ER stress. The transcription factor Xrp1 has not previously implicated in such processes. Limiting ER stress is impotent for the control of proteostasis, which is of critical importance for maintaining organism homeostasis. Loss of proteostasis is regarded as one of the major drivers of aging and an underlying cause for many pathologies including neurodegenerative diseases, some forms of diabetes and cancer.

We thank the reviewer for the high enthusiasm regarding the significance of our manuscript.

Reviewer #2:In this manuscript, the authors identified and characterized a new transcription factor that is induced by the unfolded protein response (UPR) branch PERK in *Drosophila*. They demonstrated that glutathione-S-transferases (gstD) is upregulated by the UPRactivating transgene Rh1G69D, a misfolded endoplasmic reticulum (ER) model substrate, expressed in eye discs posterior to the morphogenetic furrow. Next, they have shown that gstD is upregulated in a PERK dependent manner but independently of the other two UPR branches ATF6 and IRE1. Then, they identified that gstD is upregulated by the transcription factor Xrp1 via PERK and independently of the wellknown PERK downstream transcription factor ATF4. This is the first demonstration that Xrp1 is activated by PERK.Overall, the experiments were well designed, executed, and included detailed analysis. While most data agree with the authors' conclusions, additional experiments, and the demonstration that the PERK-Xrp1 axis is conserved in other *Drosophila* and/or other organisms would strengthen the conclusions and broaden interest in the field.Specific comments are provided below.Other major points to address1. The authors demonstrated that the overexpressed ER misfolded substrate Rh1G69D induces gstD in *Drosophila* imaginal discs. They also concluded that gstD is not induced in apical photoreceptor layer cells. Have the authors confirmed that Rh1G69D is expressed in apical photoreceptor layer cells (related to Figure 2)?

To directly address this point, we now include an anti-Rh1 labeling image in an apico-basal orientation, which shows that the photoreceptor layer also expresses Rh1 (revised Figure 2E).

2. Dithiothreitol (DTT) is commonly used to induce the UPR in unicellular tissue culture or yeast cells. Here in Figure S2, the approach to use DTT for isolated discs raised questions. Is it possible that extracellular matrix and cell-cell junctions will be affected by the DTT-induced reduction of disulfide bonds? Would it be more prudent to use tunicamycin or thapsigargin?

To address this point, we added new data in Figure S2G, H to show that tunicamycin treatment induces gstD-GFP. In addition, we added new data indicating that Xrp1 and gstD-GFP are specifically induced by *Perk* overexpression or *gadd34* RNAi (revised Figures S5, Figure 6). Since PERK is an ER protein that specifically mediates the UPR, these results strongly support the idea that gstD-GFP is induced by the PERK pathway of the UPR, and not because of artifacts associated with injuries imposed on the extracellular matrix and cell-cell junctions.

3. The Xrp1 signal is relatively dirty in the microscopy images of Figure 6A-D. It will be best if the authors complement the findings with an immunoblot that could clearly show the dependency of Xrp1 protein levels on PERK. The current microscopy images suggest a partial dependency of Xrp1 on PERK.

Anti-Xrp1 western blots from dissected imaginal discs failed to detect Xrp1, most likely because it is difficult to obtain sufficient samples through laborious dissections. To generate better images, we affinity purified the anti-Xrp1 antibody and repeated the experiment (revised Figure 6A-C). We hope the reviewer agrees that the quality of the images have improved significantly. These images now show that the number of cells inducing Xrp1 are significantly reduced.

We agree with the reviewer that there still remain small numbers of cells that still induce Xrp1. Partial dependency if Perk is common in the literature, and it is often attributed to the fact that other eIF2a kinases sometimes compensate for Perk loss of function. Based on this, we write that “While the reduction was statistically significant (Figure 6d), the residual level of Xrp1 in the Perk mutant discs indicated a partial dependency of Xrp1 induction on Perk” (revised text, page 14). To further support the role of *Perk* (and downstream eIF2a phosphorylation) in regulating Xrp1 protein levels, we added additional data in the revised manuscript showing that activating *Perk* or eIF2a phosphorylation induce Xrp1 protein levels (Figure S5, Figure 6F-J). We hope the reviewer agrees that the case for Perk regulating Xrp1 protein levels is strong.

4. How does Perk regulate Xrp1. Is it through translational inhibition as it is the case for ATF4 or by phosphorylating Xrp1 or by other intermediates/means? I was looking forward to getting the answers in the manuscript. I find it disappointing that it is left out. I strongly believe that finding the answers to these questions will strengthen the manuscript.

Thank you for the comments. In the revised manuscript, we added new data in support of the model that Perk regulates Xrp1 through phosphorylation of the translational initiation factor eIF2a, as is the case for ATF4. Specifically, we show that (1) Perk requires its kinase domain to activate the Xrp1-gstD-GFP axis (Figure S5B, C). (2) We show that conditions that increase eIF2a phosphorylation (through the knockdown of gadd34, which encodes an eIF2a phosphatase regulatory subunit) is sufficient to induce Xrp1 protein and gstDGFP expression in eye and wing imaginal discs (Figure S5D – G, Figure 6F-J). (3) We show that Xrp1 has upstream Open Reading Frames (uORFs) that are analogous to those found in ATF4. Moreover, the uORF2 coding sequence is conserved with Xrp1 coding sequences in other *Drosophila* species, which supports the idea that uORF2 is a peptide coding sequence (Figure 7).

5. It would have been great if the authors can show that Xrp1 binds directly gstD promotor. As Xrp1 is activated by PERK, are the authors predicting or have evidences that PERK-Xrp1 can upregulate other genes upon ER stress?

To address this point, we performed CUT&RUN, a modified ChIP protocol that examines Xrp1 binding sequences in vivo. We report in the revised Figure 8E that Xrp1 pull down enriches *gstD* DNA from larval tissues incubated with DTT. We further show that a known Xrp1 target, Upd3 DNA, also becomes enriched using this approach, while there is no enrichment of a negative control DNA (tubulin). These results support the idea that Xrp1 directly binds to gstD regulatory DNA in vivo.

6. In the discussion, the authors report that they could not identify any Xrp1 homologs in mammals. Are there any homologs in other species?

We now include a sentence in the Discussion (page 20), citing the conclusions from a previous study (Akdemir et al., 2007; Balnco et al., 2020) to address this point. We specifically write that “Xrp1 is well conserved in Dipteran species, but neither NCBI blast searches nor Hidden Markov Model-based analyses identify clear orthologs in other orders.” In addition, our revised Figure 7 shows Xrp1 sequence conservation between *Drosophila* species. This Figure includes an interesting observation that Xrp1’s uORF2 shows sequence conservation with Xrp1 main ORFs from other species.

7. Is the PERK-Xrp1 axis only acting in *Drosophila* eye or is the axis induce by ER stress in other tissues.

We would like to bring the reviewer’s attention to Figure S2, where we showed that ER stress induces the Xrp1 target gstD-GFP in other imaginal discs. To further drive the point, we added new data showing that knockdown of *gadd34* strongly induces Xrp1 and gstD-GFP in wing imaginal discs (Figure S5F, G, Figure gI, J).

8. Does Xrp1 upregulate other genes than gstD upon ER stress?

We would like to bring the reviewer’s attention to Figure 5E. We showed that other genes are regulated by Xrp1.

9. Although the authors elegantly dissected the PERK-Xrp1-gstD axis, the findings remain limited to *Drosophila*. I strongly believe that answering some of these questions is critical to strengthen the manuscript and to attract a larger audience.

We would like to point out that important stress response mediators such as yeast GCN4 (considered an equivalent of ATF4) and Hac1 (equivalent of XBP1) do not have clear mammalian orthologs, but nonetheless attracted broad readership. We also point out that *Drosophila* Xrp1 studies are published in reputable journals that include *eLife* (PMID 31909714). Moreover, a related manuscript describing the roles of PERK-Xrp1 in cell competition is currently under review in *eLife* (deposited to bioRxiv: https://www.biorxiv.org/content/10.1101/2021.07.12.452023v1)

Minor points to address:1. Page 10, line 22; Figure 4G should be cited after Figure 4D.

We modified the text accordingly.

2. Page 11, line 6; The induction of gstDs?

We modified the sentence.

3. Page 12, line 10-11; Is there an extra "induction" here?

Thank you for pointing this out. It has now been fixed.

Reviewer #3:This is a very solid study that identifies a novel branch of the Unfolded Protein Response. The data are of high quality and clear. The results are corroborated at multiple levels (e.g. GFP reporter and endogenous mRNA by Q-RT-PCR). The overall data package fits together into a consistent and convincing picture. The conclusions drawn from the data are well justified.

We thank the reviewer for sharing our enthusiasm.

Exceptionally, I have only very minor comments:1. Suppl. Figure 3 a-c – according to the text the clones are "negatively marked by lacZ". I assume the authors mean the clones are marked by the absence of lacZ? If so, perhaps it's easier for the reader to understand if it's written in this way.

We now modified the Supplementary Figure 3 a-c legends accordingly.

2. Suppl. Figure 3d – It would be good to include a Q-RT-PCR for cncC to show knockdown efficiency. Likewise, for Perk in main Figure 3F and crc in Figure 4D.

We now write in the revised page 9 that “the knockdown efficiency of *cnc* as estimated through q-RT-PCR from these cells was 92.46%. Likewise, we add in page 11 that “*Perk* dsRNA treatment resulted in an 82.7% reduction of *Perk* transcripts as assessed by q-RT-PCR.”

3. Suppl Figure 4a – I'm confused by the arrow. It looks like it's pointing to a cut in the eye disc? Or the arrow was moved back, leaving a black wedge overlaying the eye disc? The arrow does not seem to be pointing to a region lacking the red marker, but rather a region lacking all signal whatsoever?

We deleted the arrows, since it is only causing confusion, and as mutant clones are already marked by the absence of β-gal (red).

4. Typo Page 10 "To further test if Perk's has a cell-autonomous role". The apostrophe-s is extra.

Thank you for pointing this out. The typo has been fixed.

5. It would be helpful if the authors can speculate in the discussion how they think PERK (and presumably eIF2alpha phosphorylation) could induce XBP1 translation? Does the XBP1 mRNA have an overlapping uORF similar to ATF4?

As we wrote in our response to Reviewer 2, we now add new data indicating that Xrp1 is induced in response to gadd34 RNAi (Figure S5, Figure 6), which specifically enhances eIF2a phosphorylation. Furthermore, we show that Xrp1 mRNA has a uORF analogous to ATF4 (Figure 7). In the Discussion section (page 19), we summarize these results to propose that “Xrp1 is translationally induced, analogous to the mechanism proposed for ATF4.”

Significance:The Unfolded Protein Response (UPR) is an important stress-response pathway that plays a key role in both health and disease. The UPR has three main branches – the ATF6, the IRE1 and the PERK branches. Until now, the PERK branch is thought to signal mainly via the downstream transcription factor ATF4. By discovering an entirely novel signaling route downstream of Perk that is independent of ATF4, this broadens our understanding of the pathway, making a major contribution to our understanding of the UPR. This work will be of interest to a broad range of people working on cellular stress and homeostasis, proteotoxic stress, and associated diseases (e.g. neurodegeneration, aging, diabetes).

We very much appreciate the assessment that this work makes “a major contribution to our understanding of the UPR,” and “will be of interest to a broad range of people working on cellular stress and homeostasis, proteotoxic stress, and associated diseases.”